# The blood–brain barrier regulates brain tumor growth through the SLC36 amino acid transporter Pathetic in *Drosophila*

Qian Dong[1,2], Edel Alvarez-Ochoa[1,2], Phuong-Khanh Nguyen[1,2], Paul Orih[1,2], Natasha Fahey-Lozano[1,2], Hina Kosakamoto[3], Fumiaki Obata[3,4], Cyrille Alexandre[5], Louise Y. Cheng[1,2,6]*

1 Peter MacCallum Cancer Centre, Melbourne, Australia, 2 The Sir Peter MacCallum Department of Oncology, The University of Melbourne, Melbourne, Australia, 3 Laboratory for Nutritional Biology, RIKEN Center for Biosystems Dynamics Research, Kobe, Hyogo, Japan, 4 Laboratory of Molecular Cell Biology and Development, Graduate School of Biostudies, Kyoto University, Kyoto, Japan, 5 Epithelial Cell Interactions Laboratory, The Francis Crick Institute, London, United Kingdom, 6 Department of Anatomy and Physiology, The University of Melbourne, Melbourne, Australia

* louise.cheng@petermac.org

## Abstract

Tumors adapt their metabolism to sustain increased proliferation, rendering them particularly vulnerable to fluctuations in nutrient availability. However, the role of the tumor microenvironment in modulating sensitivity to nutrient restriction (NR) remains poorly understood. Using a *Drosophila* brain dedifferentiation neural stem cell (NSC) tumor model induced by Prospero (Pros) inhibition, we show that tumor sensitivity to NR is governed by the blood–brain barrier (BBB) glia. We found that the SLC36 amino acid transporter Pathetic (Path) regulates brain branched-chain amino acids (BCAAs) levels. Under NR, while wild-type buffers against low nutrient levels by upregulating Path, tumor glia down-regulate Path. Furthermore, Path is specifically required by the tumor (but not wildtype) BBB; its downregulation causes reduced cell cycle progression of BBB glial cells and, in turn, restricts NSC tumor growth. Path influences BBB glial cell cycle via the BCAA-mTor-S6K pathway, and its expression is controlled by Ilp6 levels and the Insulin/PI3K pathway. Overexpression of Path is sufficient to counteract the inhibitory effects of NR on tumor growth. These findings suggest that Path levels at the glial niche BBB play a key role in determining tumor sensitivity to NR.

## Introduction

Altered metabolism is a hallmark of cancer [1]. In particular, disruptions in amino acid metabolism have been shown to selectively inhibit tumor growth based on their genetic profile and tissue of origin [2–4]. Identifying the metabolic vulnerabilities linked to these cancer-specific characteristics offers opportunities for novel dietary

**Data availability statement:** All relevant data are within the paper and its Supporting Information files.

**Funding:** QD is supported by the Peter MacCallum Cancer Foundation Grant (2319). FO's lab is supported by Japan society for the promotion of science (JPJSBP120249944), LYC's laboratory is supported the NHMRC Ideas Grant (APP2011289). The funders had no role in study design, data collection and analysis, decision to publish, or preparation of the manuscript.

**Competing interests:** The authors have declared that no competing interests exist.

**Abbreviations:** ALH, after larval hatching; BBB, blood–brain barrier; BCAAs, branched-chain amino acids; BCAT, branched-chain amino acid transaminase; CDD, chemically defined diet; Cdk, cyclin-dependent kinase; CG, cortex glial; CNS, central nervous system; CW, critical weight; FC, Fold Change; FDR, false discovery rate; FOXO, Forkhead box O; GLAD, Gene List Annotation for *Drosophila*; GMC, ganglion mother cell; Ilps, Insulin-like peptides; IPCs, Insulin-producing cells; LC–MS/MS, Liquid Chromatography–Tandem Mass Spectrometry; NBs, neuroblasts; NL, neural lamella; NR, nutrient restriction; NSC, neural stem cell; PBS, phosphate-buffered saline; PG, perineural glia; SG, salivary gland; SPG, sub-perineural glia; TCA, Tricarboxylic Acid; WD, wing discs.

and therapeutic interventions targeting cancer growth. In brain tumors, moderate dietary restriction has been shown to significantly reduce astrocytoma growth [5], while cysteine and methionine withdrawal sensitize gliomas to ferroptosis [6]. Additionally, elevated glucose levels are associated with increased proliferation and poor drug response in glioblastoma cell lines [7,8]. Despite the therapeutic potential of dietary interventions in brain cancer, the mechanisms linking diet to brain tumor growth remain unclear, highlighting the need for further investigation into the relationship between metabolism and tumor progression.

In this study, we utilize a *Drosophila* brain tumor model induced via dedifferentiation, a central mechanism underlying tumor formation, to study the effect of nutrient restriction (NR) on tumorigenesis. *Drosophila* has a simple central nervous system (CNS) where neural stem cells (NSCs), called neuroblasts (NBs), undergo asymmetric divisions to produce diverse neuronal and glial populations that form adult fly brains (Fig 1A). During brain development, NBs reside in a specialized microenvironment, or niche, which provides essential signals and trophic support to regulate NB behavior [9,10]. This niche consists of cortex glial (CG) cells, which form individual chambers to house each NB and its progeny [11–15]. Additionally, the surface glial cells, including the perineural glia (PG) and sub-perineural glia (SPG), collectively form the blood–brain barrier (BBB) at the brain surface, which is covered by a sheath of extracellular matrix known as the neural lamella (NL, Fig 1A) [16–18]. This barrier prevents the passage of drugs or toxins while facilitating the uptake of nutrients via amino acid, glucose, and lipid transporters [17,19–22].

Type I NBs are the most common NB type found throughout the CNS (Fig 1B). They produce neurons by first generating an intermediate progenitor cell called ganglion mother cell (GMC), where the homeodomain transcription factor Prospero (Pros) promotes its differentiation into two postmitotic neurons [23]. We previously showed that NB lineages can buffer against nutrient shortage, and the ability to do so is determined by the activation of the PI3K signaling pathway, via the glial niche secreted ligand Jelly Belly (Jeb) and NB-specific Anaplastic Lymphoma Kinase [24–26]. Here, we found that the brain tumors caused by the loss of Pros (Fig 1C) are in fact sensitive to NR [24]. This sensitivity is attributed to the downregulation of the SLC36A4 amino acid transporter, Pathetic (Path) under NR, which leads to reduced BBB glial cell cycle progression and, in turn, tumor growth. In contrast, Path is upregulated in control brains upon NR and is not required under fed conditions. Path influences BBB glial number via the mTor-S6K pathway, and its expression is controlled by Ilp6 levels and the insulin/PI3K pathway. Notably, overexpressing Path is sufficient to counteract the inhibitory effects of NR on tumor growth. Path has been shown to transport amino acids such as tryptophan, proline, alanine *in vitro*, however, we did not find levels of these AA to be altered upon Path knockdown. Instead, we found Path knockdown affected the levels of brain BCAAs levels which are critical for glial expansion and tumor growth. These findings suggest that Path levels at the BBB play a key role in determining the response of brain tumors to NR.

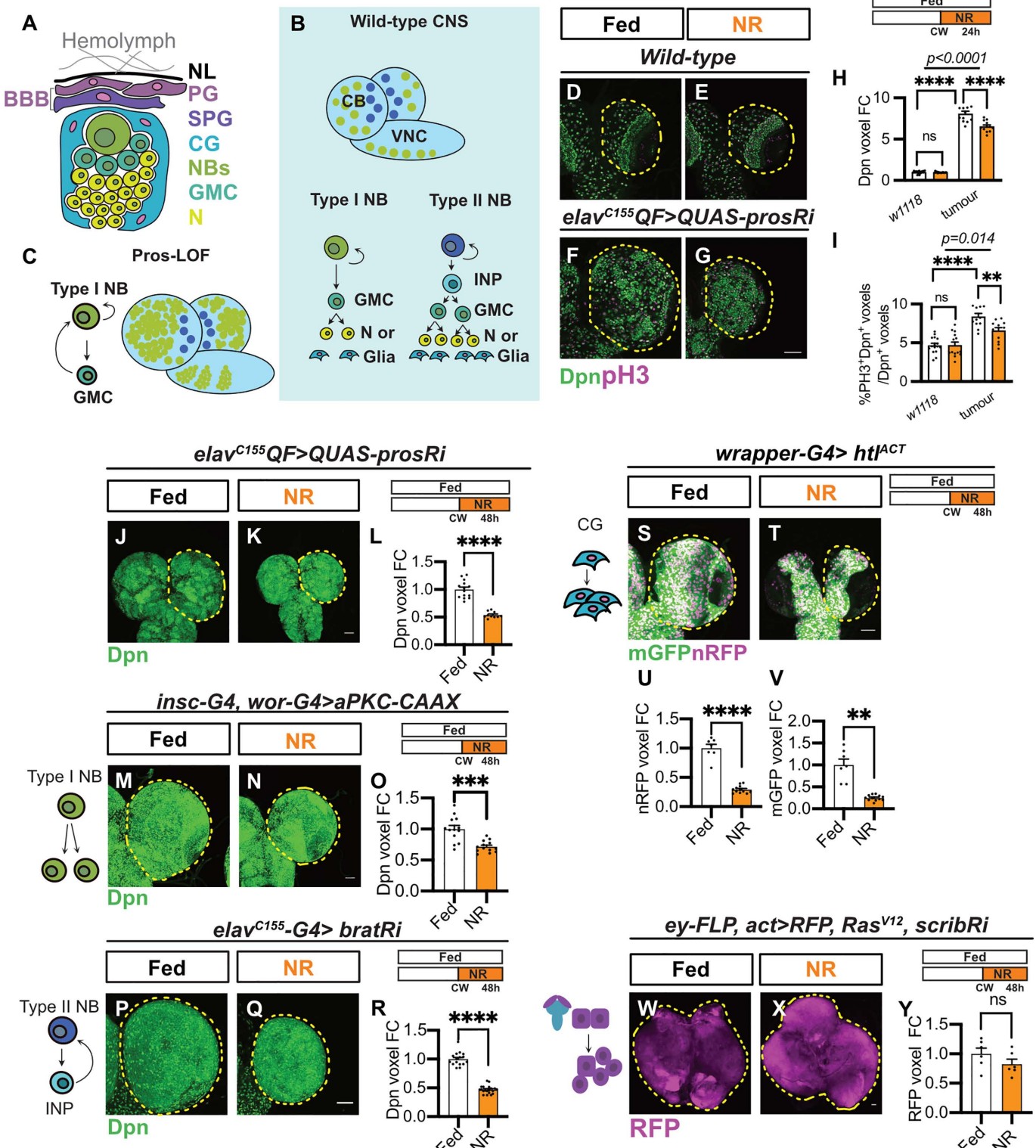

**Fig 1. The growth of brain tumors is sensitive to post-critical weight (CW) nutrient restriction (NR). (A)** Schematic representation of the NB niche, which is composed of perineural glia (PG) and sub-perineural glia (SPG), forming the blood–brain barrier (BBB) at the brain surface under neural lamina (NL); and cortex glia (CG), which directly encase NBs, ganglion mother cells (GMCs), and neurons (N). **(B)** Schematic representation of the larval central nervous system (CNS). The CNS consists of two brain lobes and one ventral nerve cord (VNC). Type I neuroblasts (NBs, green), in the central

brain (CB) region of brain lobes and VNC, undergo asymmetric divisions to produce a self-renewing NB and a GMC, which further divides once to make two differentiated neurons or glia (yellow). Type II NBs (blue) are limited to the dorsal side of the CB. They divide asymmetrically and generate an intermediate neural progenitor (INP), which undergoes several rounds of divisions. **(C)** Schematic depicting the formation of Prospero (Pros) Loss-of-function (LOF) NB tumors. Loss of the transcription factor Pros causes the reversion of GMCs to type I NBs (green), which undergo unlimited divisions to form NB tumors. **(D-G)** single-section images of *w1118* brain lobes **(D, E)** and *elav$^{C155}$QF>QUAS-prosRi* tumor brains **(F, G)** stained with Deadpan (Dpn) and the mitotic marker pH3 under Fed and NR conditions. CBs are circled by yellow dashed lines. NR: 72–96hALH; Dissection: 96hALH. **(H)** Quantification of the normalized (to *w1118*_Fed) Dpn voxels in **(D–G)** (*n* = 16, 15,12,12). **(I)** Quantification of the percentage of type I NBs undergoing mitosis (pH3$^+$Dpn$^+$) in each brain lobe in **(D–G)** (*n* = 16, 15,12,12). **(J, K)** Maximum projection images of *elav$^{C155}$QF>QUAS-prosRi* tumor brains stained with Dpn under Fed and NR conditions. NR: 72–120hALH. **(L)** Quantification of normalized (to Fed) Dpn voxels in **(J, K)** (*n* = 14, 10). **(M, N)** Overexpression of *aPKC-CAAX* with the NB driver *insc-G4, wor-G4* induces symmetric division of type I NB and formation of NB tumors. (M, N) are maximum projection images of tumor brain lobes marked by Dpn under Fed and NR conditions. **(O)** Quantification of normalized (to Fed) Dpn voxels of circled brain lobes in **(M, N)** (*n* = 13, 13). **(P, Q)** Overexpression of *bratRNAi* with the NB driver *elav$^{C155}$-G4* causing reversion of INPs to type II NBs, forming NB tumors in the dorsal side of the CB. (P, Q) are maximum projection images of tumor brain lobes marked by Dpn under Fed and NR conditions. **(R)** Quantification of normalized (to Fed) Dpn voxels of circled brain lobes in **(P, Q)** (*n* = 16, 20). **(S, T)** Overexpression of *htl$^{ACT}$* with the CG driver *wrapper-G4* causes over-proliferation of CG, **(S, T)** are maximum projection images of these brain lobes marked with mGFP and nRFP under Fed and NR conditions. **(U, V)** Quantification of normalized (to Fed) nRFP voxels and mGFP voxels of circled brain lobes in **(S, T)**, respectively (*n* = 7, 12). **(W, X)** Co-expression of *Ras$^{V12}$* and *scribRNAi* in eye discs (*ey-FLP, act-G4*) leads to the formation of neoplastic tumors (marked by *UAS-RFP* and circled in W, X). **(W, X)** are maximum projection images of eye disc tumors under Fed and NR conditions. **(Y)** Quantification of normalized (to Fed) RFP voxels in **(W, X)** (*n* = 6, 6). Data information: ALH, after larvae hatching. NR: 72–120hALH; Dissection: 120hALH unless otherwise stated. Scale bar = 50 μm. Error bar represents SEM. In **(H)**: Two-way ANOVA was used to analyze whether the effect of NR on Dpn volume is different in wildtype vs. the *elavQF>QUAS-prosRi* tumor-bearing brains (significance indicated by interaction *P* value (*p* < 0.0001)). Mean ± SEM, and statistical results including multiple comparisons are displayed in S1A Table. In **(I)**: Two-way ANOVA was used to analyze whether the effect of NR on NB proliferation is different in wildtype vs. the *elavQF>QUAS-prosRi* tumor-bearing brains (significance indicated by interaction *P* value (*p* < 0.0001)). Mean ± SEM, and statistical results including multiple comparisons are displayed in S1B Table. In **(L)**: Welch's *t* test, (****) *P* < 0.0001. In **(O)**: Welch's *t* test, (***) *P* = 0.0003. In **(R)**: Welch's *t* test, (****) *P* < 0.0001. In **(U)**: Welch's *t* test, (****) *P* < 0.0001. In **(V)**: Welch's *t* test, (**) *P* = 0.0013. In **(Y)**: unpaired *t* test, (ns) *P* = 0.2013. Raw data are included in S3 Data.

## Results

### The growth of dedifferentiation-induced NB tumors is sensitive to NR

We previously demonstrated that neurogenesis becomes insensitive to dietary restriction after critical weight ([CW], around 60 hours after larval hatching [ALH]), a developmental checkpoint beyond which starvation no longer impacts the initiation of metamorphosis or survival [25–27]. Consistent with this, in wild-type brains, we showed that both NB number and the percentage of NBs undergoing mitosis remained unaffected, following 24 hours of NR post-CW (NR: complete starvation on 0.4–1% agar, Figs 1D, 1E, 1H, 1I, and S1A–S1D). Next, we assessed whether brain tumors would respond differently to NR. We induced dedifferentiation-derived NB tumors by expressing *QUAS-prosRNAi* using the pan-NB lineage driver, *elav$^{C155}$-QF2*, and subjected the animals to NR starting from CW. Brain tumors in *Drosophila* induced by *pros* knockdown cause developmental delays due to the disruption of ecdysone signaling (as shown by [28]). To investigate the relationship between tumor growth and NR, we first identified when tumor-bearing larvae reached CW. By subjecting these larvae to NR at various developmental stages, we assessed the temporal sensitivity of tumors to nutrient availability (S1E Fig). We determined that tumor-bearing *Drosophila* larvae reached CW at approximately 68 hours ALH. Larvae subjected to NR before this time point exhibited severe developmental consequences: those starved from 48 hours ALH died, and those starved from 63.5 hours ALH failed to pupariate on time (S1F Fig). In comparison, subjecting the animals to NR from 68 hours ALH and onwards caused them to pupariate earlier (S1F Fig) [29]. NR effectively suppressed the growth of most polyploid tissues in tumor-bearing animals, including the fat body and salivary glands (SGs) (S1G–S1H and S1J–S1K Fig), similar to its effects in wild-type animals [25]. Unlike wild-type brains, brain tumors exhibited significant growth reduction following 24 or 48 hours of NR post-CW compared to fed conditions (Figs 1D–1L, S1I, and S1L). After 24-hour NR, we observed a significant reduction in the total number of NBs (marked by Dpn) in tumor but not wild-type brains (Fig 1D–1H). The effect of NR on brain tumor growth is more pronounced after 48-hour NR (Fig 1J–1L). From this point onward, tumor-bearing animals were subjected to NR after CW for either 24 hours if we are comparing with the wildtype wildtype animals reach wandering stages 24 hours after CW) or 48 hours, when comparing between tumor

genotypes, unless otherwise specified. These findings highlight the unique sensitivity of brain tumors to NR, even when wild-type brain proliferation is unaffected.

To elucidate the mechanisms accounting for the reduced tumor growth, we examined tumor proliferation, differentiation, and cell death. The reduction in brain tumor size following NR was found to be due to a slowdown in cell cycle progression. This was evidenced by a decrease in the incorporation of the thymidine analogue EdU (5-ethynyl-2′-deoxyuridine) into S phase of the cell cycle (15-minute, S2A–S2C Fig), and the percentage of tumor cells undergoing mitosis (marked by pH3, Fig 1F, 1G, and 1I). As the tumor arises from dedifferentiation, we next assessed whether the tumor size reduction upon NR is due to increased differentiation from NBs into neurons. We observed a slight decrease in the percentage of neurons (Elav/DAPI ratio, S2D–S2F Fig), indicating that NR reduces tumor growth not by inducing more conversion of NBs to neurons. The reduction in brain tumor size following NR was also not accounted for by increased cell death, as indicated by the unaltered percentage of NBs marked by the apoptotic marker Dcp-1 (S2G–S2I Fig). This suggests that NR impedes tumor growth primarily by slowing down cell cycle progression, rather than by promoting differentiation or inducing apoptosis.

We next assessed whether NR affects the growth of different tumor types, including other brain and epithelial tumors. We found that NR significantly reduced the size of brain tumors induced in (1) type I NBs by the constitutive activation of aPKC (Fig 1M–1O) [30]; (2) type II NBs by the knockdown of Brat (Fig 1P–1R) [31]; and (3) CG cells, by the constitutive activation of the FGF receptor Heartless (Fig 1S–1V) [32]. In all cases, tumor size was significantly reduced after 48 hours of NR (Fig 1M–1V). These findings demonstrate that NR broadly impacts tumor growth across various brain tumor models. In contrast, the growth of the eye-antennal disc tumors, induced via the activation of Ras ($Ras^{V12}$) and the knockdown of Scribble (scrib RNAi) [33], was not significantly affected by NR (Fig 1W–1Y). Therefore, NR sensitivity could represent a distinctive characteristic of brain tumors, distinguishing them from wild-type tissue and epithelial tumors.

## The BBB glial expansion is significantly reduced under NR in tumor brains

The brain tumor lineages are surrounded by a glial niche made up of CG, PG, and SPG (Fig 1A) [9]. This glial niche has been shown to provide the trophic support necessary for wild-type neural proliferation [9]. These glial cells likely play a key role in modulating the tumor microenvironment and supporting tumor growth, which may contribute to the observed sensitivity of brain tumors to NR. To test this, we first examined how the number of total glial cells, marked by the pan-glial nuclear marker Reversed Polarity (Repo), is altered under NR conditions. We found that wild-type brains exhibited a mild glial number reduction of 13.22% after 24 hours of NR (from an average of 461–400, refer to S3 Data for all raw counts, Fig 2A, 2B, and 2G) [34], and the tumor glial cell number was reduced by 19.2% after 24 hours of NR (from 927 to 749, S3 Data) and 47.78% after 48 hours of NR compared to Fed (from 1,485 to 776, S3 Data and Fig 2C–2F and 2G–2H). This suggests NR has a greater effect on glial numbers in tumor brains than in wild-type brains. Similarly, we found that bratRNAi and aPKC-CAAX brain tumors also exhibited significant reductions in total glial cell number upon 48 hours of NR compared to Fed (S3A–S3F Fig), suggesting that reduction in glial cell number under NR maybe a common feature of brain tumors.

Next, we examined how the subsets of glial cells are affected. The BBB, composed of PG and SPG, regulates nutrient transport into the brain. During development, PG actively divide through mitosis and expand in number as the CNS grows [16,32]. In comparison, SPG undergo endoreplication and endomitosis to increase cell size proportional to brain growth and maintain BBB junctional continuity [16–19]. We found that the number of SPG cells (marked by moody-GAL4>Stinger, S3G–S3J Fig), and CG cells (marked by wrapper-GAL4>UAS-nRFP) were significantly reduced (S3K–S3M Fig) compared to Fed. To validate our findings using an alternative driver, we characterized the expression of GMR85G01-GAL4, which has been shown to be expressed in surface glial cells in the adult CNS [35]. We found that GMR85G01-GAL4 is expressed in glial cells marked by Repo staining at the brain surface (S4B–S4B‴ and S4D Fig), but did not colocalize with Repo or Elav staining in the deep layers of the CNS (S4A–S4A‴, S4C–S4C‴, and S4D Fig). Together, our data indicates

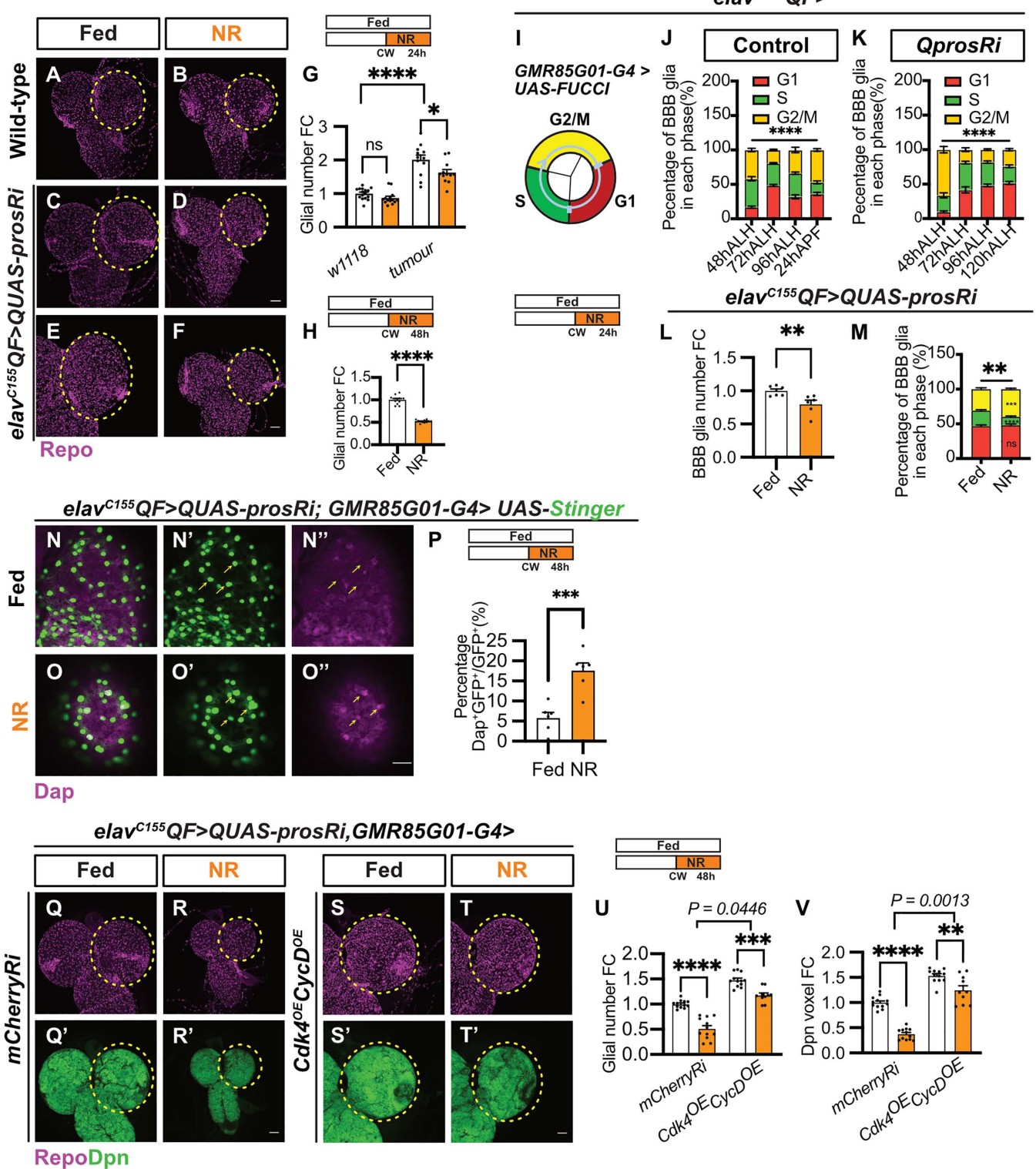

**Fig 2. Blood–brain barrier (BBB) glial cell cycle progression is compromised under nutrient restriction (NR), and this underlies the high susceptibility of brain tumors to NR. (A–D)** Maximum projection images of wildtype and *elav^C155QF>QUAS-prosRi* tumor brains stained with Repo under Fed vs. NR. NR: 72-96hALH; Dissection: 96hALH. **(E, F)** Maximum projection images of *elav^C155QF>QUAS-prosRi* tumor brains stained with Repo under

Fed vs. NR. NR: 72-120hALH; Dissection: 120hALH. **(G)** Quantification of normalized (to wildtype_fed) glial number in the brain lobe in **(A–D)** ($n = 16$, 14, 12, 12). **(H)** Quantification of normalized (to Fed) glial number in the brain lobe in **(E, F)** ($n = 11$, 12). **(I)** Schematic depicting Fly-FUCCI, a tool that enables the visualization of cell cycle transitions by tagging E2F1 and Cyclin B proteins with ECFP (red) and Venus (green). **(J, K)** Quantification of the percentage of BBB glial cells in each cell cycle phase within the circled control and tumor brain lobes at different time points (images are in S4I–S4J′″ Fig) ($n = 10$, 7, 7, 4, 8, 5, 6, 4). **(L, M)** Quantification of the total number of BBB glia **(L)** and the percentage of BBB glial cells in each cell cycle phase **(M)**, labeled by FUCCI using *GMR85G01-G4* in *elav^C155^QF>QUAS-prosRi* tumor brain lobes under Fed vs. NR ($n = 7$, 6). NR: 72-96hALH; Dissection: 96hALH. **(N, O)** Single-section images of *elav^C155^QF>QUAS-prosRi* tumor brain lobes where BBB glia are marked with *GMR85G01-G4> UAS-Stinger*, and Dap expression shown in magenta (yellow arrows), under Fed and NR conditions. NR: 72-120hALH; Dissection: 120hALH. Scale bar: 20 µm. **(P)** Quantification of the percentage of BBB glia expressing Dap in each lobe in **(N, O)**. **(Q–T)** Maximum projection images of *elav^C155^QF>QUAS-prosRi* tumor brains where *Cdk4* and *CycD* were co-expressed in BBB glia using *GMR85G01-G4* compared with *mCherryRi* under Fed vs. NR. Glia: Repo, and NBs: Dpn. NR: 72-120hALH; Dissection: 120hALH. **(U, V)** Quantifications of the normalized (to *mCherryRi* fed condition) glial number **(U)** and Dpn voxels **(V)** of each circled brain lobe in **(Q–T)** ($n = 12$, 12, 12, 10). Data information: ALH, after larvae hatching; APF, after pupa formation. Brain lobes are circled with yellow dashed lines. Scale bar = 50 µm. Error bar represents SEM. In **(G)**: Two-way ANOVA was used to analyze the effect of NR on glial number in wildtype vs. the *elavQF>QUAS-prosRi* tumor bearing brains. Mean ± SEM, and statistical results including multiple comparisons are displayed in S1C Table. In **(H)**: Welch's *t* test, (****) $P < 0.0001$. In **(I)**: unpaired *t* test, (**) $P = 0.0072$. In **(J, K)**: Two-way ANOVA were used to analyze the distribution of BBB glial cells in each cell cycle phase over time in both wild-type brains and tumor-bearing brains. In **(J)**: interaction $P < 0.0001$. In **(K)**: interaction $P < 0.0001$. Mean ± SEM, and statistical results including multiple comparisons are displayed in S1D and S1E Table. In **(L)**: unpaired *t* test, (**) $P = 0.0073$. In **(M)**: Chi-squared test, (**) $P = 0.0014$. Mean ± SEM, and statistical results, including Two-way ANOVA multiple comparisons, are displayed in S1F Table. In **(P)**: unpaired *t* test, (***) $P = 0.0006$. In **(U, V)**: Two-way ANOVA were used to analyze whether the effect of NR on glial number and tumor size is affected by *CdK4* and *CycD* overexpression (significance indicated by interaction *P* value). In **(U)**: interaction $P = 0.0446$; In **(V)**: interaction $P = 0.0013$. Mean ± SEM, and statistical results including multiple comparisons are displayed in S1G and S1H Table. Raw data are included in S3 Data.

*GMR85G01-GAL4* is a robust BBB glial driver. Besides its expression in the CNS, this driver is also expressed in larval gut, SG, and wing discs (WD), but not the fat body (FB) (S4E–S4H Fig).

By expressing the fly-FUCCI [36] using *GMR85G01-GAL4* (Fig 2I), we showed that BBB glial cells actively progressed through the cell cycle during early larval stages and continued to proliferate post-CW in wild-type and tumor brains (Figs 2J, 2K and S4I–S4J′″). In tumor-bearing brains, BBB glial cell cycle slowed progressively over time, as indicated by an increase in the percentage of G1-phase cells (Figs 2K and S4J–S4J′″). After 24 hours of NR, the number of BBB glial cells was reduced (Fig 2L), and this is accounted for by the reduction of cells in S phase and increase in G2/M phase (Fig 2M). To assess whether this is caused by cell cycle arrest, we examined the expression of cyclin-dependent kinase (Cdk) inhibitor, Dacapo (Dap, [37]). We found a 3-fold increase in Dap⁺ BBB glial cells under NR compared to Fed (Fig 2N–2P). Increased G2/M cell cycle arrest is typically associated with increased cell death [38], however, we did not observe a significant change in glial cell death following NR (S4K–S4M Fig). These findings suggest that NR inhibits BBB glial cell cycle progression, and may account for tumor reduction.

To test whether enhancing BBB glial cell cycle progression could promote tumor growth under NR, we overexpressed the cell cycle genes *Cdk4* and *CycD* using the *GMR85G01-GAL4* driver. Compared to the control (*mCherryRNAi*), the overexpression of *Cdk4* and *CycD* increased overall glial number under fed conditions (from 1,184 to 1,749 in average, S3 Data and Fig 2Q–2T and 2U). Furthermore, this manipulation partially rescued the decrease in glial numbers caused by NR (Fig 2Q–2T and 2U, the number of glia is reduced by 48% upon NR in *mCherryRi*, and 20% in *Cdk4^OE^, CycD^OE^*, statistical assessment described in Materials and methods). As a result, the decrease in tumor size upon NR was also partially rescued in the *Cdk4/CycD*-overexpressing condition compared to *mCherryRNAi* control (Fig 2Q′–2T′ and 2V, the size of tumor is reduced by 63% upon NR in *mCherryRi*, and 18% in *Cdk4^OE^, CycD^OE^*, statistical assessment described in Materials and methods). These findings suggest that manipulating BBB glial cell numbers can influence the sensitivity of brain tumor growth to NR, highlighting the role of BBB glia in tumor response to nutrient availability.

## Branched-chain amino acids (BCAAs) are required for glial and tumor expansion

During development, growth is fueled by nutrients derived from the diet, including carbohydrates and proteins. To identify the essential nutrient components for glial and tumor expansion, we selectively removed yeast (protein source) or glucose

and polenta (sugar source) from the standard diet post CW. We then assessed the impact of these nutrient dropouts on glial and tumor growth, aiming to determine which nutrients are critical for supporting the expansion of glial cells and tumor progression. We found that the withdrawal of yeast reduced glial number and tumor growth, phenocopying the effects of NR (Fig 3A–3G). To test the sufficiency of yeast or glucose in promoting tumor growth, we supplemented the diet with 2× yeast or 2× glucose post CW. We found that neither of these manipulations is sufficient to increase glial cell numbers or tumor size (Fig 3A–3G).

Amino acids constitute a significant component of yeast. Using Liquid Chromatography–Tandem Mass Spectrometry (LC–MS/MS), we investigated the impact of NR on amino acid (AA) levels. Our analysis revealed a substantial reduction in AA levels in both the hemolymph and the tumor brains (S5A and S5B Fig). AAs are imported into the brain through transporters located on its surface. To further understand how NR affects the brain's nutrient uptake, we examined the expression of nutrient transporters in tumor brains under NR conditions. We compared the expression of annotated transporters (using Gene List Annotation for *Drosophila* [GLAD]) in tumor brains under Fed and NR conditions via bulk RNA-sequencing. We demonstrated that the gene expression profile of tumor-bearing brains under NR diverges from that of the Fed (S6A and S6B Fig), with 225 genes significantly upregulated and 301 significantly downregulated (False Discovery Rate [FDR] ≤0.05, Fold Change [FC] ≥ 1.5, S1 Data). We found that AA transporters, including Minidiscs (MND), Juvenile hormone Inducible-21 (JhI-21), and Pathetic (Path), were among the top transcriptionally downregulated transporters in the tumor brain under NR (Fig 3H). Furthermore, we observed a reduction in the carbohydrate transporter Major Facilitator Superfamily Transporter 3, the lipid transporter fatty acid binding protein, and several ATP synthetase components and subunits (S2 Data).

MND and JhI-21 are reported to be transporters for branched-chain amino acids (BCAAs), i.e., leucine (Leu), isoleucine (Ile), and valine (Val) [39–41]. MND is not expressed in the BBB, while JhI-21 and Path are expressed there [17,42]. Given that both *mnd* and *JhI-21* were transcriptionally downregulated upon NR, we next tested whether BCAAs are rate-limiting for the expansion of glia and tumors, using a chemically defined diet (CDD) [43,44]. We showed that the removal of Leu or Ile from 72 hours ALH resulted in a reduction in glial cell number and tumor size (Fig 3I–3N). Brain tumors are known to be addicted to dietary methionine (Met) [45]. However, in our study, the withdrawal of Met did not significantly affect glial cell number or tumor growth (Fig 3I–3N). Our data indicate that BCAAs are crucial dietary components for glial and tumor growth. To further investigate their role, we analyzed the cell cycle profiles of BBB glial cells under Leu or Ile dropout conditions. We found that the reduction in glial numbers was driven by a slowdown in BBB glial cell cycle progression, as evidenced by a significant increase in the percentage of cells in the G1-phase and a significant reduction in the number of cells in the S-phase (Fig 3O). In comparison, these changes were not observed under Met dropout (Fig 3O). These results suggest that BCAAs play a key role in regulating BBB glial expansion and, consequently, tumor growth.

We next examined how deprivation of BCAAs slows down tumor growth. BCAAs, particularly Leu, are known to support cell growth by promoting protein synthesis and by fueling the Tricarboxylic Acid (TCA) cycle [46]. In line with this, our RNA-seq data showed that the Aminoacyl-tRNA biosynthesis and the TCA cycle are among the top 10 downregulated pathways under NR compared to fed conditions (S6C Fig). We found that leucyl-tRNA synthetase, isoleucyl-tRNA synthetase, but not valyl-tRNA synthetase, which function to conjugate the amino acids to the corresponding tRNA for protein synthesis at the ribosome, are significantly downregulated under NR (S6D Fig). However, enzymes including branched-chain amino acid transaminase (BCAT) and branched-chain alpha-ketoacid dehydrogenase complexes (Bckdha, Bckdhb, and Dbct, identified and classified in flies in [47]), which convert BCAAs to the TCA cycle, are not altered under NR (S6D Fig). These indicate that downregulation of BCAA tRNA synthetases might account for why NB tumor growth is compromised under NR. Indeed, temporal knockdown of leucyl-tRNA synthetase in the NB tumor (*elav^C155^-GAL4> prosRi*) dramatically reduced the tumor size and the percentage of tumor cells that are pH3+(pH3 index, Fig 3P–3S), suggesting that leucyl-tRNA synthetase is necessary for NB tumor growth. Similarly, this manipulation also slows down wild-type brain growth (DAPI staining, S6E–S6G Fig), suggesting leucyl-tRNA synthetase is similarly required for wild-type brain

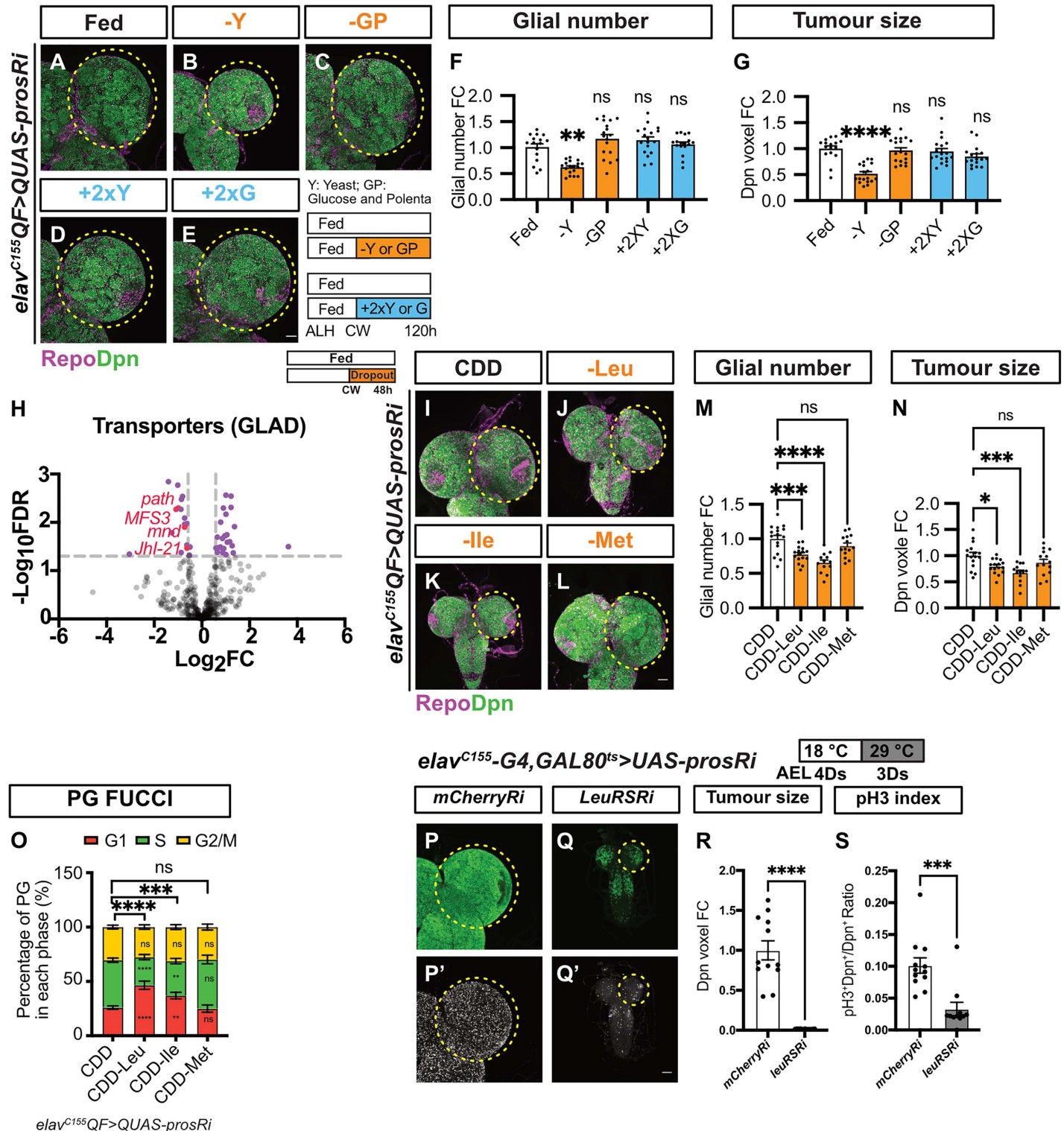

**Fig 3. Branched-chain amino acids (BCAAs) are necessary for glial and neuroblast (NB) tumor expansion. (A–E)** Maximum projection images of *elav^{C155}QF>QUAS-prosRi* tumor brains stained with Repo and Dpn, under Fed **(A)**, dropout of Yeast **(B)**, glucose and polenta **(C)**, 2× Yeast supplementation **(D)** or 2× Glucose supplementation **(E)** post-CW from 72-120hALH. **(F, G)** Quantifications of the normalized (to Fed) glial number **(F)** and Dpn voxels

(G) of each circled brain lobe in (A–E) ($n = 16, 18, 18, 18, 17$). (H) Volcano Plot depicting differential gene expression in the $elav^{C155}QF>QUAS\text{-}prosRi$ tumor brains under NR compared to Fed  Genes that are significantly altered (False discovery rate [FDR] < 0.05; FC > 1.5) are marked with purple dots. NR: 72-120hALH; Dissection: 120hALH. (I–L) Maximum projection images of $elav^{C155}QF>QUAS\text{-}prosRi$ tumor brains stained with Repo and Dpn. The animals were moved from standard food to CDD (I), CDD-Leu (J), CDD-Ile (K), and CDD-Met (L) at 72hALH and dissected at 120hALH. CDD, chemically defined diet. (M, N) Quantifications of the normalized (to Fed) glial number (M) and Dpn voxels (N) of each circled brain lobe in (I–L) ($n = 16, 16, 13, 14$). (O) Quantifications of BBB glial cell cycle distribution in $elav^{C155}QF>QUAS\text{-}prosRi$ tumor brain lobes, where FUCCI was overexpressed using $GMR85G01\text{-}G4$, under fed (CDD), dropout of Leu, Ile, or Met ($n = 10, 5, 8, 6$). (P, Q) Maximum projection images of $elav^{C155}\text{-}G4$, $GAL80^{ts}>UAS\text{-}prosRi$ tumor brains with $mCherryRi$ or $LeuRSRi$ overexpression. Larvae were placed at 18 °C for 4 days before being moved to 29 °C for transgene activation (3 days). (P, and Q): Dpn; (P' and Q'): phosphorylated histone H3 (pH3). (R, S) Quantifications of the normalized (to $mCherryRi$) Dpn voxels (R) and pH3 index (S) of each circled brain lobe in (P, Q) ($n = 12,11$). Data information: ALH, after larvae hatching. Brain lobes are circled with yellow dashed lines. Scale bar = 50 μm. Error bar represents SEM. In (F): Kruskal–Wallis test, (**) $P = 0.0021$; (ns) $P = 0.6207$; (ns) $P = 0.8285$; (ns) $P > 0.9999$. In (G): Kruskal–Wallis test, (****) $P < 0.0001$; (ns) $P > 0.9999$; (ns) $P > 0.9999$; (ns) $P = 0.2290$. In (M): One-way ANOVA, (***) $P = 0.0006$; (****) $P < 0.0001$; (ns) $P = 0.2437$. In (N): One-way ANOVA, (*) $P = 0.0189$; (***) $P = 0.0002$; (ns) $P = 0.2308$. In (O), the Chi-squared test was used to compare the difference in cell cycle phase distribution between nutritional conditions. (****) $P < 0.0001$; (***) $P = 0.0001$; (ns) $P = 0.6442$. Two-way ANOVA was used to compare the differences in each cell cycle phase between nutritional conditions. Mean ± SEM, and statistical results, including multiple comparisons, are displayed in S1I Table. In (R): Welch's $t$ test, (****) $P < 0.0001$. In (S): Mann–Whitney test, (***) $P = 0.0004$. Raw data are included in S3 Data.

development. However, we found that in wild-type animals, post-CW NR did not change the expression level of this enzyme (S6H Fig). This partially accounts for why the wild-type brains are spared under post-CW NR. Collectively, our data suggest that dietary BCAAs are required for brain tumor growth via two mechanisms: first, they regulate the expansion of BBB glial cells, which controls the overall nutrients including BCAAs that can be transported into the brain; and second, they promote protein synthesis of NB tumors to support their growth.

## The amino acid transporter Pathetic (Path) is downregulated in the BBB of tumor brains under NR

RNA sequencing data indicated that the expression of the SLC36 amino acid transporter Pathetic (Path) is significantly downregulated in tumor brains under NR. To further investigate this, we examined changes in Path protein expression using a Path-GFP protein trap [48]. Consistent with previous studies, we observed that Path-GFP was expressed in wild-type glial cells, marked by *repo-GAL4>mRFP* (S7A Fig) [26] and NB tumors marked by Mira (S7G and S7H Fig). We found that Path-GFP levels was not significantly altered in the BBB between the wild-type and tumor brains (S7A–S7C Fig). Under NR, in wild-type brains, Path-GFP levels were elevated in the BBB glia (S7D–S7F Fig). In contrast, Path-GFP expression in the BBB glia of tumor brains was significantly reduced compared to Fed conditions (Fig 4A–4D). This finding suggests that NR induces differential regulation of Path expression in glial cells depending on the presence of tumors. In addition, we observed a reduction in Path-GFP expression in both CG cells (the glia that form the immediate niche surrounding NB tumors) and tumor NBs under NR (S7G–S7I Fig). Interestingly, NR-resistant $Ras^{V12}scrib^{RNAi}$ tumor exhibited no significant change to the Path expression level in response to NR (S7J–S7L Fig). Collectively, these findings suggest that Path expression may correlate with tumor sensitivity to NR.

## Path is necessary and sufficient for BBB glial expansion and tumor growth

It was previously shown that Path is required for NB proliferation [8]. To assess whether Path is required for glial expansion, we knocked down *path* in tumor glial cells (*repo-GAL4*) using *pathRNAi* (KK). This resulted in a reduction of both total glial numbers and tumor size (S8A–S8D Fig), and these effects were further recapitulated using the BBB glial driver (*GMR85G01-GAL4*, Fig 4E–4H), but not a CG driver (*NP2222-GAL4* [49], S8H–S8K Fig). Induction of *pathRNAi* (KK) in BBB glia using *GMR85G01-GAL4* eliminated Path-GFP expression at the BBB (marked by *UAS-mRFP*, S8E–S8G Fig), validating the effectiveness of the reagent. Similar effects on glial number and tumor size were found using an independent *pathRNAi* line with *dcr2* driven by *repo-GAL4* (S8L–S8O Fig). Together, these findings suggest that Path is required in the BBB to regulate glial expansion and tumor growth.

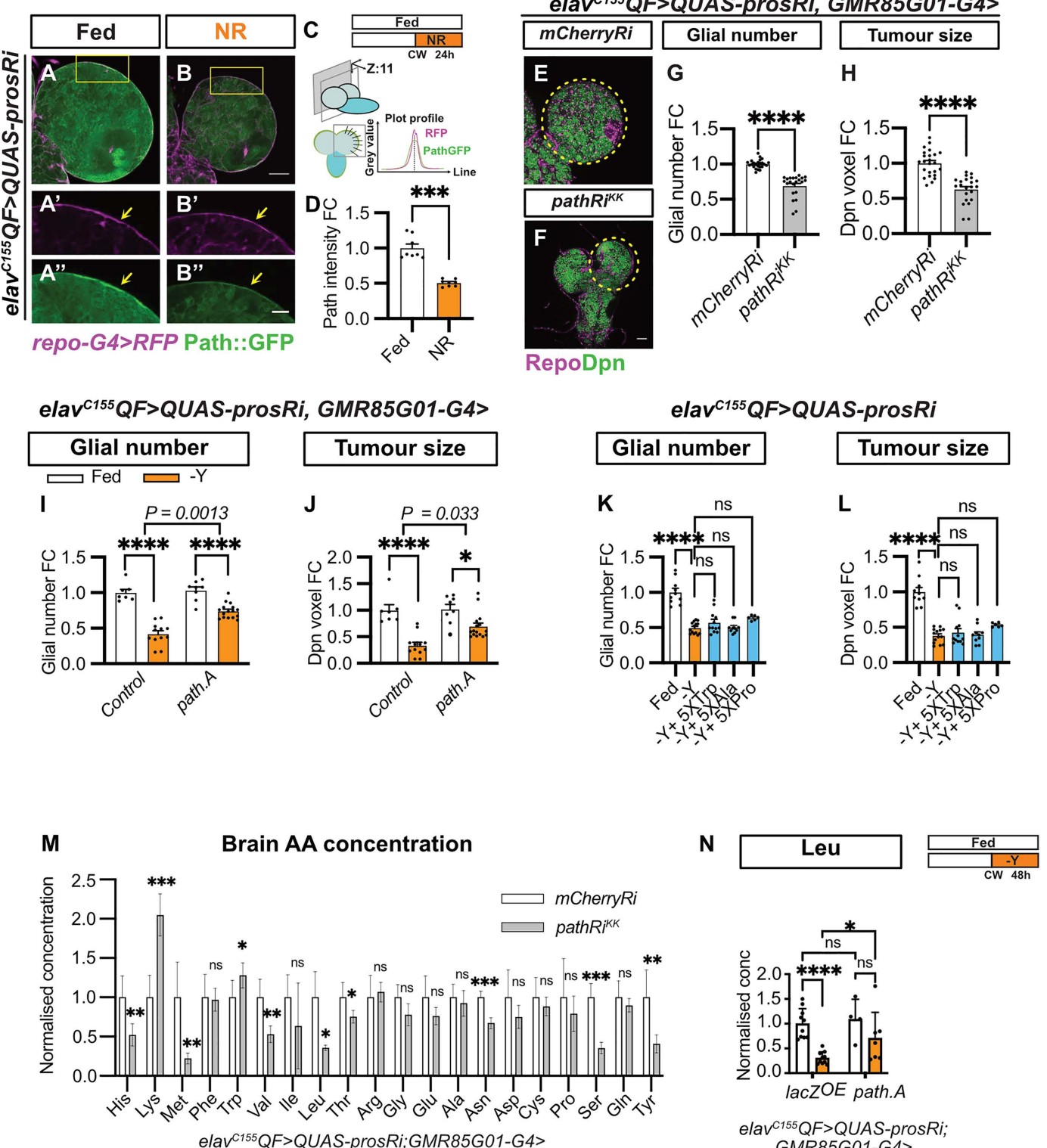

**Fig 4. Pathetic (Path) is downregulated at the blood–brain barrier (BBB) under nutrient restriction (NR), restricting the availability of essential amino acids (EAAs) for tumor growth. (A–B)** Single-section **(A–B)** and zoomed-in images **(A′–B″)** of *elav^{C155}QF>QUAS-prosRi* tumor brains where Path-GFP is expressed in the glia (marked by *repo-G4>mRFP*) at the brain surface and is downregulated under NR compared to Fed conditions. NR: 72–96hALH; Dissection: 96hALH. Scale bar = 20 μm in (A′–B″). **(C)** Schematic depicting Path-GFP intensity quantification (described in Materials

and methods). **(D)** Quantification of the normalized (to Fed) Path-GFP intensity at the brain surface in (A, B) (n = 8, 8). **(E, F)** Single-section images of *elav^C155^QF>QUAS-prosRi* tumor brains with *pathRi^KK^* **(F)** overexpressed in the BBB using *GMR85G01-G4*, compared to *mCherryRi* **(E)** at 120hALH. Glia: Repo; NBs: Dpn. **(G, H)** Quantifications of the normalized (to *mCherryRi*) glial number **(G)** and Dpn voxels **(H)** of each circled brain lobe in **(E, F)** (n = 24, 22). **(I, J)** Quantifications of the normalized (to control Fed condition) glial number (I) and Dpn voxels **(J)** of *elav^C155^QF>QUAS-prosRi* tumor brain lobes, where *path.A* was overexpressed in the BBB glia using *GMR85G01-G4*, compared with *control* (GMR85G01-G4 >w1118) under Fed vs.−Yeast (n = 7, 12, 8, 6). Yeast dropout: 72-120hALH; Dissection: 120hALH. **(K, L)** Quantifications of the normalized (to Fed) glial number **(K)** and Dpn voxels **(L)** of *elav^C155^QF>QUAS-prosRi* tumor brain lobes under Fed, dropout of Yeast (−Y) and −Yeast with 5xTrp or 5xAla or 5xPro conditions from 72-120hALH (n = 10, 14, 12, 10, 6). **(M)** Quantifications of the normalized (to *mCherryRi*) AA concentration in *elav^C155^QF>QUAS-prosRi* tumor brains, where *mCherryRi* or *pathRi^KK^* was overexpressed in the BBB glia using *GMR85G01-G4*. **(N)** Quantifications of the normalized (to lacZOE_Fed) brain Leu concentration in *elav^C155^QF>QUAS-prosRi* tumor-bearing animals, where *lacZ* or path is overexpressed using *GMR85G01-G4* under Fed and −Yeast conditions (n = 10, 10, 4, 7). Raw data are included in S3 Data. Data information: ALH, hours after larvae hatching. Brain lobes are circled with yellow dashed lines. Scale bar = 50 μm. Error bar represents SEM. In **(D)**: Mann–Whitney test, (***) $P = 0.0002$. In **(G)**: Mann–Whitney test, (****) $P < 0.0001$. In **(H)**: Mann–Whitney test, (****) $P < 0.0001$. In **(I, J)**: Two-way ANOVA were used to analyze whether the effect of NR on glial number or tumor size is affected by *path.A* overexpression in BBB glia (significance indicated by interaction P value). In **(I)**: interaction $P = 0.0013$. In **(J)**: interaction $P = 0.0330$; Mean ± SEM, and statistical results including multiple comparisons are displayed in S1J and S1K Table. In **(K)**: One-way ANOVA, (****) $P < 0.0001$, (ns) $P = 0.4251$, (ns) $P > 0.9999$, (ns) $P = 0.0964$. In **(L)**: Kruskal–Wallis test, (****) $P < 0.0001$, (ns) $P > 0.9999$, (ns) $P > 0.9999$, (ns) $P = 0.1124$. In **(M)**: His: unpaired t test, (**) $P = 0.0082$; Lys: unpaired t test, (***) $P = 0.0003$; Met: unpaired t test, (**) $P = 0.0049$; Trp: unpaired t test, (*) $P = 0.0480$; Val: unpaired t test, (**) $P = 0.0032$; Leu: Welch's t test, (*) $P = 0.0110$; Thr: unpaired t test, (*) $P = 0.0266$; Asn: unpaired t test, (***) $P = 0.0001$; Ser: unpaired t test, (****) $P < 0.0001$; Tyr: unpaired t test, (**) $P = 0.0068$. In **(N)**: Two-way ANOVA, *lacZ^OE^_fed vs. path.A_fed*: (ns) $P = 0.6666$, *lacZ^OE^_fed vs. lacZ^OE^_-Y*: (****) $P < 0.0001$, *pathA_fed vs. pathA_-Y*: (ns) $P = 0.0832$, *lacZ^OE^_-Y vs. path.A_-Y*: (*) $P = 0.0230$.

Importantly, the overexpression of *path* (*UAS-path.A* [50]) in the BBB glia was able to partially override the effect of NR on glial and tumor growth. Compared to the control, this manipulation partially rescued the decrease in glial number and tumor size under starvation (yeast dropout) (Fig 4I and 4J, the number of glia and tumor size are reduced by 58% and 67%, respectively, upon yeast dropout in control, and 28% and 32% in *path.A*, statistical assessment described in Materials and methods). Together, these results suggest that Path is required by the BBB in tumor brains and its overexpression can partially enhance glial proliferation under NR. To assess whether Path is generally required in regulating gliogenesis, we knocked down *path* using *repo-GAL4* in wild-type brains. We found that *path* knockdown did not affect glial numbers in wild-type brains under fed conditions (S8P–S8R Fig). However, *path* knockdown reduced glial number and NB proliferation under NR (S8S–S8V Fig), consistent with previous findings [26]. Together, these data suggest that while wild-type glia does not need Path, glia growth under stress (such as tumor or under NR) are dependent on Path.

## Path regulates the availability of amino acids in the brain

Path has been shown to have high affinity, but low capacity to transport its substrates tryptophan (Trp), proline (Pro), or alanine (Ala) [51,52]. We found that supplementing the yeast-free food with 5xTrp, Pro or Ala from 72 to 120 hours ALH was insufficient to restore glial or tumor growth (Fig 4K and 4L), suggesting that Path is unlikely to affect tumor growth through these AAs. To assess if AA levels in the brain is affected by *path* knockdown in BBB, we conducted tumor brain LC–MS/MS. We found that *path* knockdown in the BBB glia led to a reduction in BCAAs Leu and Val, as well as other AAs such as His, Met, Thr, Ser, Asn, and Tyr (Fig 4M), but no significant changes to Trp, Pro, or Ala. Interestingly, we observed an increase in Lys level. It is possible that *path* knockdown might up-regulate the expression of transporters such CAT [53], that can mediate Lys transport. It is also possible that Lysine catabolism (by LKRSDH) is downregulated, which can lead to an increase of Lysine levels [54]. *path* overexpression in BBB glia is sufficient to partially restore brain Leu levels (Fig 4N) and other AAs including Asn and Tyr (S9 Fig). Together, our data suggest Path can regulate BCAA in the brain, however, we could not distinguish between the following possibilities: (1) BCAA is directly transported by Path; (2) BCAA could be affected by Trp, Pro or Ala levels that are transported by Path; and (3) changes in Path may cause compensatory regulation of other AA transporters such as LAT that can also alter the transport of BCAAs.

**Path expression is regulated by Ilp6-InR/PI3K pathway**

Next, we investigated the regulatory signals influencing Path expression. First, we evaluated the role of the PI3K pathway, a well-established mediator of cellular growth in response to nutrient availability [55]. We first assessed whether the PI3K pathway is altered in the glia of tumor brains under NR by using a Forkhead box O (FOXO)-GFP reporter. FOXO is known to translocate from the cytoplasm to the nucleus when the PI3K pathway is inhibited (Fig 5A) [56]. Under NR, we found that FOXO level was increased in the nuclei of BBB glia cells in tumor brains (assessed at superficial brain sections, Fig 5B–5D), suggesting that the PI3K pathway is downregulated under NR. Next, we investigated whether altering the InR/PI3K pathway influences Path expression. Expression of a constitutively active form of *InR* (*InR$^{CA}$*) significantly increased Path-GFP expression in the BBB (Fig 5E, 5F, and 5H). Given that this manipulation delayed larval development, we further validated our result with *repo-GAL4*, where activation of *InR$^{CA}$* consistently increased Path-GFP expression in glia (S10A Fig). Conversely, the expression of a dominant-negative form of the Insulin Receptor (*InR$^{DN}$*) in BBB glia (using *GMR85GO1-GAL4*) significantly reduced Path-GFP levels (marked by mRFP) (Fig 5E, 5G, and 5H). Similar results were obtained using *repo-GAL4*, and in addition to *InR$^{DN}$*, overexpression of the PI3K adaptor protein Dp60 also reduced Path-GFP expression in glia (S10A Fig). Together, our data indicate that Path expression is positively regulated by the InR/PI3K pathway. Furthermore, we observed a mild decrease in the total Foxo-GFP level in glial nuclei (S10B–S10D Fig), suggesting that Path lies downstream of InR/PI3K signaling. Functionally, temporal inhibition of the PI3K pathway, achieved by overexpressing Dp60 in the glia of tumor brains post-CW (Fig 5I), significantly decreased glial numbers and tumor size (Fig 5J–5M), mimicking the effects of NR. And overexpressing Path in the BBB glia led to a partial rescue of both glial number and tumor size, which are reduced by *InR$^{DN}$* expression (Fig 5N–5R). Additionally, temporal activation of the PI3K effector AKT (*myrAKT*) resulted in a smaller reduction in both glial number and tumor size under NR compared to Fed (Fig 5S–5U, the number of glia and tumor size are reduced by 73% and 77%, respectively, upon NR in *mCherryRi*, and by 25% and 25% in *myrAKT$^{OE}$*, statistical assessment described in Materials and methods). Together, our findings suggest that NR downregulates PI3K signaling and in turn *path* expression, resulting in reduced glial number and tumor size.

To understand how insulin signaling is regulated by NR, we assessed the source of Insulin-like peptides (Ilps), which are known to activate insulin signaling. Insulin-producing cells (IPCs) are known to be the main source of Ilps, which regulate systemic growth [57,58]. We previously showed that the expression of *p60* in the IPCs leads to a reduction in IPC size, consequently reducing the release of Ilps into the hemolymph [25]. Similarly, this manipulation in tumor-bearing animals resulted in smaller pupae (S10E, S10F, and S10G Fig) and halted the growth of peripheral tissues (e.g., SGs, S10E′, S10F′, and S10H Fig). However, this unexpectedly led to an increase in glial number and tumor size, in contrast to the effects observed with NR (S10E″, S10F″, S10I, and S10J Fig). Our data suggests that systemic Ilps reduction is not mediating the effect of NR on glial and tumor growth.

Other sources of Ilps include adipose tissue (the fat body) [59,60] and glial cells within the CNS [32,61]. We found that Ilp6 is expressed at lower levels in the fat body and glial cells of tumor-bearing animals compared to controls (examined via *Ilp6-GAL4>nGFP* [62]) (Figs 6A–6J and S10K). Under NR, while fat body Ilp6 levels became elevated in control animals (Fig 6A–6B′ and 6E) [59,62], they remained unaltered in tumor-bearing animals (Fig 6C–6D′ and 6E). Conversely, Ilp6 expression in glia, especially in CG but not BBB glia, was reduced upon NR in both control and tumor animals (Figs 6F–6J and S10K). To determine the functional importance of Ilp6 in the fat body and glial cells, we inhibited Ilp6 in these tissues post-CW (Fig 5I). Inhibiting Ilp6 in the fat body led to a significant reduction in both glial number and tumor size (Fig 6K–6N). Similarly, expressing *Ilp6 RNAi* in glia post-CW mildly reduced glial number and tumor size (Fig 6O–6R). These findings demonstrate that both fat body- and glial-derived Ilp6 play important roles to maintain insulin signaling in the BBB glia to maintain its proliferation and in turn support tumor expansion.

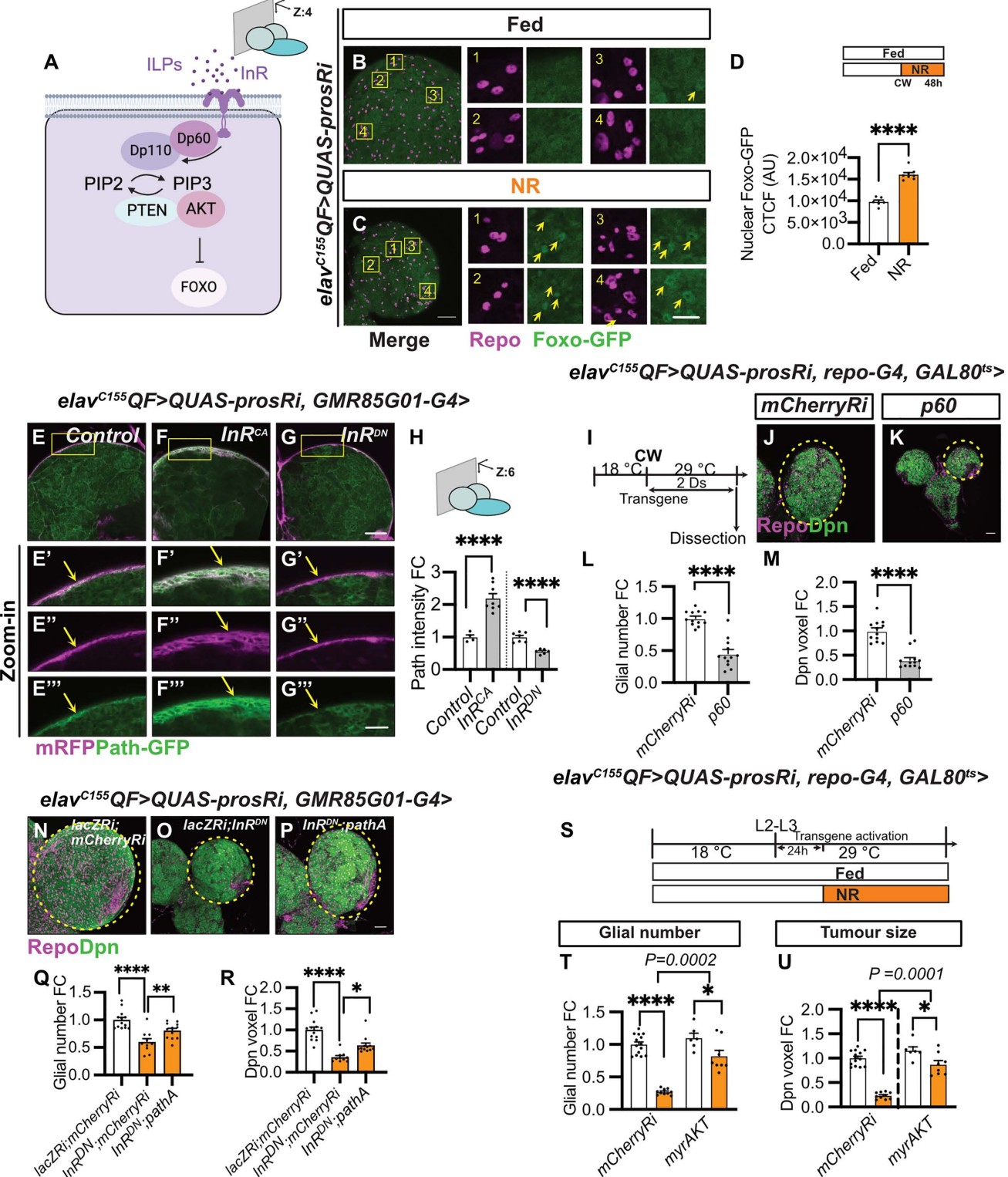

**Fig 5. Glial Path expression is regulated by the InR/PI3K pathway. (A)** Schematic depicting the PI3K pathway. Created in BioRender. Dong, Q. (2025) https://BioRender.com/l19t132. **(B, C)** Surface-section images and zoomed-in images of Foxo-GFP expression (Yellow arrows) in *elav^C-155^QF>QUAS-prosRi* tumor brains with glia stained with Repo under Fed and NR. NR: 72–120hALH; Dissection: 120hALH. Scale bar = 20 μm in

zoomed-in images (1–4). **(D)** Quantification of glial nuclear Foxo-GFP intensity in (B, C, glia nucleus: Repo) ($n = 6, 8$). **(E–G)** deep-section **(E–G)** and zoomed-in images (E′–G‴) of Path-GFP expression (yellow arrows) in *elav$^{C155}$QF>QUAS-prosRi* tumor brains with *InR$^{CA}$* or *InR$^{DN}$* overexpressed in BBB glia. The BBB is marked with *GMR85G01-G4>mRFP*. **(H)** Quantification of the normalized (to *control*) Path-GFP intensity at the BBB in (E–G) ($n = 4, 8,$ 7, 6). The control in the first column is the same as that in S7C Fig. Scale bar = 20 μm in (E′–G‴). **(I)** Schematic depicting the transgene activation regime in **(J, K)**. Tumor-bearing animals were moved from repressive 18 °C to permissive 29 °C 6–7 days after egg laying (AEL) to allow transgene activation for 2 days from CW onwards. **(J, K)** Single-section images of *elav$^{C155}$QF>QUAS-prosRi* tumor brains with *p60* (K) overexpressed in glia using *repo-G4* according to the regime in (I), compared to *mCherryRi* (J). Glia: Repo; NBs: Dpn. **(L, M)** Quantifications of the normalized (to *mCherryRi*) glial number **(L)** and Dpn voxels **(M)** of each circled brain lobe in (J, K) ($n = 12, 12$). The same *mCherryRi* data was used in **(L)** and Fig 6Q; The same *mCherryRi* data was used in **(M)** and Fig 6R. **(N–P)** maximum projection images of *elav$^{C155}$QF>QUAS-prosRi* brains, where *lacZRi;mCherryRi*, *lacZRi;InR$^{DN}$* or *InR$^{DN}$;pathA* is overexpressed using *GMR85G01-G4*, Glia: Repo; NBs: Dpn. Dissection: 120hALH. **(Q, R)** Quantifications of the normalized (to *lacZRi;m-CherryRi*) glial number **(Q)** and Dpn voxels **(R)** of each circled brain lobe in (N–P) ($n = 13, 10, 12$). **(S)** Schematic depicting the transgene activation and NR regime in **(T, U)**. Genotype: *elav$^{C155}$QF>QUAS-prosRi, repo-G4, GAL80$^{ts}$>mCherryRi* vs. *myrAKT* in **(T, U)**. Tumor-bearing animals were moved from 18 to 29 °C at the L2–L3 transition to allow transgene activation only from L3 onwards. Animals were placed on either standard food or 0.42% Agar/PBS for NR 24 hours after L3 (equivalent to the timing of CW at 25 °C) for 2 days before dissection. **(T, U)** Quantifications of the normalized (to *mCherryRi* fed condition) glial number **(T)** and Dpn voxels **(U)** of *elav$^{C155}$QF>QUAS-prosRi* tumor brain lobes, where *myrAKT* were overexpressed in glia according to **(S)**, compared with *mCherryRi* under Fed vs. NR ($n = 14, 10, 6, 8$). Data information: ALH, hours after larvae hatching. Brain lobes are circled with yellow dashed lines. Scale bar = 50 μm. Error bar represents SEM. In **(D)**: unpaired *t* test, (****) $P < 0.0001$. In **(H)**: One-way ANOVA, (****) $P < 0.0001$; unpaired *t* test, (****) $P < 0.0001$. In **(L)**: One-way ANOVA, (****) $P < 0.0001$. In **(M)**: Kruskal–Wallis test, (****) $P < 0.0001$. In **(Q)**: One-way ANOVA, (****) $P < 0.0001$, (**) $P = 0.0069$. In **(R)**: Kruskal–Wallis test, (****) $P < 0.0001$, (*) $P = 0.0338$. In **(T, U)**: Two-way ANOVA were used to analyze whether the effect of NR on glial number and tumor size is affected by *myrAKT* overexpression in glia (significance indicated by interaction $P$ value). In **(T)**: interaction $P = 0.000$. In **(U)**: interaction $P = 0.0001$. Mean ± SEM, and statistical results including multiple comparisons are displayed in S1N and S1O Table. Raw data are included in S3 Data.

## Path regulates BBB glial expansion and tumor growth via mTOR/S6K pathway

Path/SLC36A4 transporter has been associated with growth regulation through the mTOR/S6K signaling pathway (Fig 7A) in *Drosophila* wings and mice retinal pigmented epithelium [52,63,64]. To determine whether this relationship extends to glial cells in brain tumors, we overexpressed the dominant negative forms of S6K (via *S6K$^{KQ}$*) and mTOR (*Tor$^{TED}$*) in tumor brains using *repo-GAL4*. We found that these manipulations resulted in reductions in glial number and tumor size (Fig 7B–7G). We next assessed whether activation of mTOR/S6K signaling pathway in BBB glia is sufficient to rescue the decrease in glial number and tumor size induced by NR. We showed that temporal activation of S6K (*S6K$^{CA}$*) or Rag (*RagA$^{CA}$*) in BBB glia were sufficient to induce a smaller reduction in glial number and tumor size upon NR compared to the *mCherryRi* control (Fig 7H–7J, the number of glia and tumor size are reduced by 54% and 65%, respectively, upon NR in *mCherryRi*, by 20% and 30% in *RagA$^{CA}$*, and by 38% and 36% in *S6K$^{CA}$*, statistical assessment described in Materials and methods). Finally, while the activation of S6K did not change total glial number and tumor size, it resulted in a rescue of both parameters upon *path* knockdown (Fig 7K and 7L). Furthermore, inhibition of mTOR (*Tor$^{TED}$*) using *repo-GAL4* did not significantly affect Path-GFP expression in the surface glial of tumor brains (Fig 7M–7O), suggesting that mTOR likely functions downstream of Path to regulate glial proliferation and tumor growth. However, we were not able to assess the effect of Path knockdown on mTOR and S6K activity due to technical limitations (The antibody against pS6K showed very low expression in glial cells). Collectively, our data suggest Path regulates BBB glial expansion potentially via the mTOR-S6K pathway, and its expression level is regulated by the fat body and glia-derived Ilp6 and the InR/PI3K pathway.

## Discussion

Upon the transformation of normal cells into cancer cells, metabolic rewiring occurs to meet the increased proliferative demands of the tumor [1,65,66]. Limiting nutrient availability in the microenvironment impacts cell growth and proliferation in both physiological and pathological contexts. However, strategies targeting the altered metabolism of tumors remain limited [67], partly because different tumors and cell types within the same tumor exhibit distinct metabolic vulnerabilities [68]. Understanding cell-type-specific metabolic dependencies within a tumor is therefore critical for developing effective therapeutic approaches.

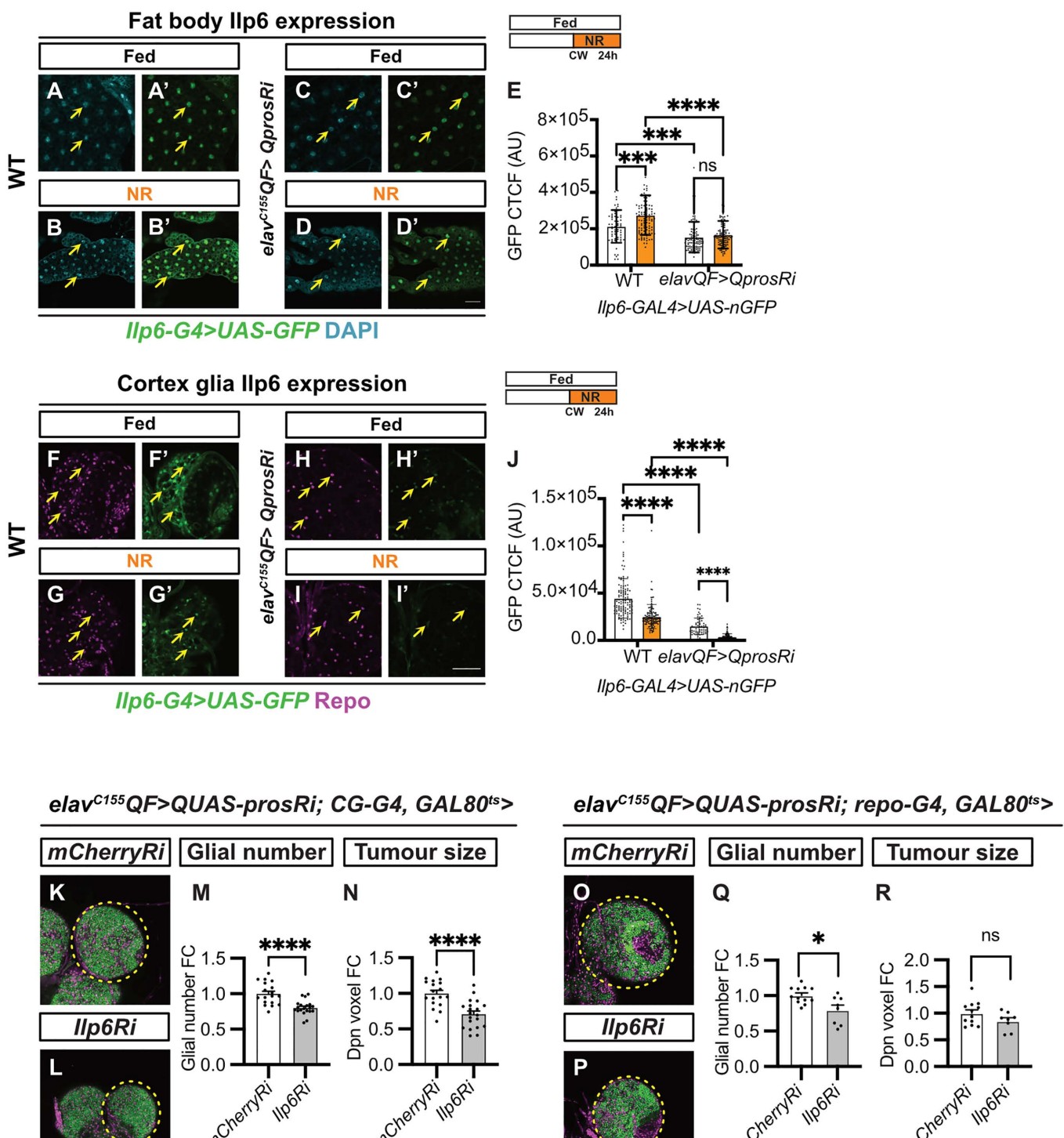

**Fig 6. The glial PI3K pathway is sustained by Ilp6 secreted from the fat body and glia. (A–D′)** Single-section images of Ilp6 expression (*Ilp6-G4>nGFP*) in fat body (FB) of wild-type and *elav^{C155}QF>QUAS-prosRi* tumor brains under fed and NR conditions (yellow arrows). Wild-type and tumor-bearing animals were starved after CW (65hALH and 68hALH, respectively). The cell nucleus was marked with DAPI. **(E)** Quantifications of GFP

intensity (CTCF) in FB nucleus in (A–D′) (*n* = 63, 104, 102, 99 cells from 7, 9, 9, 9 FBs). (**F–I′**) Single-section images of Ilp6 expression (*Ilp6-G4>nGFP*) in CG of wild-type and *elav^C155^QF>QUAS-prosRi* tumor brains under Fed and NR conditions (yellow arrows). Wild-type and tumor-bearing animals were starved after CW (65hALH and 68hALH, respectively). CG cells were marked with Repo and distinguished from other glial cell types based on location. (**J**) Quantifications of GFP intensity (CTCF) in CG nucleus in (**F–I′**) (*n* = 120, 115, 60, 96 cells from 12, 12, 6, 10 brain lobes). (**K, L**) Single section images of *elav^C155^QF>QUAS-prosRi* tumor brains with *Ilp6Ri* (**L**) overexpressed specifically in the fat body using *CG-G4* according to the regime in (Fig 5J), compared to *mCherryRi* (**K**). Glia: Repo; NBs: Dpn. (**M, N**) Quantifications of the normalized (to *mCherryRi*) glial number (**M**) and Dpn voxels (**N**) of each circled brain lobe in (**K, L**) (*n* = 18, 20). (**O, P**) Single section images of *elav^C155^QF>QUAS-prosRi* tumor brains with *Ilp6Ri* (**P**) overexpressed in glia using *repo-G4* after CW according to the regime in (Fig 5J), compared to *mCherryRi* (**O**). Glia: Repo; NBs: Dpn. (**Q, R**) Quantifications of the normalized (to *mCherryRi*) glial number (**Q**) and Dpn voxels (**R**) of each circled brain lobe in (**O, P**) (*n* = 12, 8). Data information: ALH, after larvae hatching. Brain lobes are circled with yellow dashed lines. Scale bar = 50 μm. Error bar represents SEM. In (E): Two-way ANOVA were used to analyze whether the effect of NR on FB Ilp6 expression is different in the tumor brains compared with wild-type brains (significance indicated by interaction *P* value). Interaction *P* = 0.0129. Multiple comparisons: WT fed vs. NR: (***) *P* = 0.0001; Tumor fed vs. NR: (ns) *P* = 0.7166; WT fed vs. Tumor fed: (***) *P* = 0.0003; WT NR vs. Tumor NR: (****) *P* < 0.0001. In (J): Two-way ANOVA were used to analyze whether the effect of NR on CG Ilp6 expression is different in the tumor brains compared with wild-type brains (significance indicated by interaction *P* value). Interaction *P* = 0.0070. Multiple comparisons: WT fed vs. NR: (****) *P* < 0.0001; Tumor fed vs. NR: (****) *P* < 0.0001; WT fed vs. Tumor fed: (****) *P* < 0.0001; WT NR vs. Tumor NR: (****) *P* < 0.0001. In (M): unpaired *t* test, (****) *P* < 0.0001. In (N): unpaired *t* test, (****) *P* < 0.0001. In (Q): One-way ANOVA, (*) *P* = 0.0437. In (R): Kruskal–Wallis test, (ns) *P* = 0.7729. Raw data are included in S3 Data.

This study leverages the QF-QUAS and GAL4-UAS binary expression systems to investigate the role of nutrients in modulating brain tumor growth, where we were able to genetically manipulate key regulators in the BBB (using the GAL4-UAS system) and assess the effects on the tumors (generated by the QF-QUAS system). We demonstrate that NR reduces brain tumor growth by downregulating the expression of the SLC36A4 amino acid transporter Path specifically in the BBB of tumor-bearing brains (Fig 7P). Under normal conditions, Path expression at the BBB of tumor brains is similar to that of its wild-type counterparts; however, NR triggers the downregulation of Path exclusively in tumor brains, leading to tumor growth inhibition. Functionally, Path is both necessary and sufficient to sustain BBB glial cell cycle progression, and its expression can override the inhibitory effects of NR on tumor growth. Moreover, Path expression is regulated by InR/PI3K signaling, sustained by two distinct pools of Ilp6 derived from the fat body and glial cells. In wild-type animals, the fat body upregulates *Ilp6* under NR; however, in tumor-bearing animals, this upregulation fails to occur. Previous studies have demonstrated that 20E is sufficient to induce *Ilp6* expression [59]. We have also shown that tumor-bearing animals exhibit a deficit in systemic 20E levels [28]. Therefore, it is likely that the absence of 20E prevents the upregulation of *Ilp6* under NR conditions in tumor brains, a hypothesis we will test in follow-up studies.

In this study, we have identified the BBB as a critical nutrient transportation hub for modulating the response of NB tumors to NR. We have also found that the growth of these BBB glial cells is directly controlled by the amino acid transporter Path. While Path has been shown to regulate overall body size and NB proliferation in developmental contexts, it is not required for the growth of other tissues, such as oenocytes, fat body, and SGs [26,50,52]. In tumor contexts, Path was previously shown to be upregulated to enhance the import of AAs, thereby promoting the growth of WD tumors [63]. Our results extend this finding, supporting a growth-promoting role for Path in the BBB of tumor brains, highlighting its essential function in tumor growth regulation under nutrient-restricted conditions.

Despite previous findings suggesting that Path is essential for scavenging proline to drive tumor growth in high-sugar-dependent tumorigenesis [63], our study did not identify proline as the primary substrate for Path in this context. Unlike other SLC36 transporters, which typically function by coupling AA transport with proton exchange [69], SLC36A4/Path operates at neutral pH and exhibits a high affinity but low capacity for transporting AA substrates like Trp, Pro, and Ala [51,52,70]. In line with this, we show that supplementing animals with Trp, Pro, or Ala was not able to increase glial number or tumor size under NR. Therefore, these AAs are unlikely to be the rate-limiting substrates for Path in driving brain tumor growth. Instead, inhibition of Path in the BBB of tumor brains led to a reduction in the concentration of BCAAs, suggesting that BCAAs may play a more critical role in the regulation of tumor growth, likely through mechanisms yet to be fully elucidated. We showed that downstream of Path is the mTOR-S6K pathway. It has been proposed that Path functions as a transceptor which can regulate signaling pathways independent of its ability to transport amino acids [50].

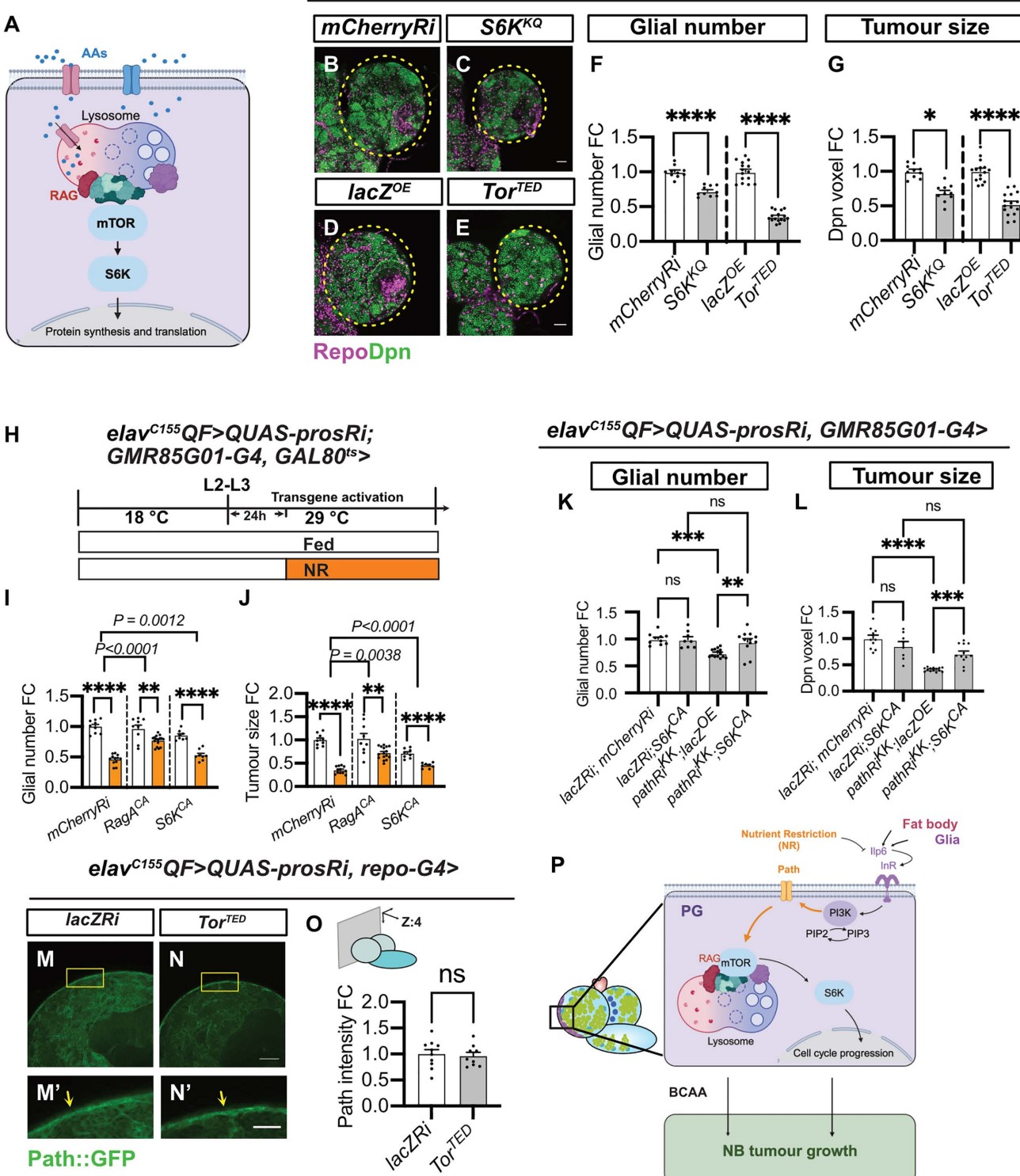

**Fig 7. Path regulates glial proliferation via the mTOR-S6K pathway. (A)** Schematic depicting the activation of the mTOR-S6K pathway by AAs through Rag GTPases. Created in BioRender. Dong, Q. (2025) https://BioRender.com/a373zo9. **(B–E)** Single section images of *elav^C-155^QF>QUAS-prosRi* tumor brains, where *mCherryRi, S6K^KQ^, lacZ,* and *Tor^TED^* were overexpressed in glia using *repo-G4*. Glia: Repo; NBs: Dpn. **(F, G)**

Quantifications of the normalized (to *mCherryRi* or *lacZ^OE*) glial number **(F)** and Dpn voxels **(G)** of each circled brain lobe in **(B–E)** (*n* = 10, 10, 15, 16). **(H)** Schematic depicting the transgene activation and NR regime in **(I, J)**. Genotype: *elav^C155^QF>QUAS-prosRi, GMR85G01-G4, GAL80^ts^> mCherryRi* vs. *Rag^CA^* vs. *S6K^CA^* in **(I, J)**. Tumor-bearing animals were moved from repressive 18 °C to permissive 29 °C at the L2–L3 transition to allow transgene activation from L3 onwards. Animals were placed on either standard food or 0.42% Agar/PBS for NR 24 hours after L3 (equivalent to the timing of CW at 25 °C) for 2 days before dissection. **(I, J)** Quantifications of the normalized (to *mCherryRi* Fed condition) glial number **(I)** and Dpn voxels **(J)** of *elav^C155^QF>QUAS-prosRi* tumor brain lobes, where *Rag^CA^* or *S6K^CA^* were overexpressed in the BBB glia according to **(H)**, compared with *mCherryRi* under Fed vs. NR (*n* = 9, 12, 9, 16, 8, 8). **(K, L)** Quantification of normalized (to *lacZRi; mCherryRi*) glial number **(K)** and Dpn voxels **(L)** of each *elav^C155^QF>QUAS-prosRi* tumor brain lobe, where *lacZRi;S6K^CA^, pathRi^KK^;lacZ^OE^*, and *pathRi^KK^;S6K^CA^* were overexpressed in BBB glia, compared with *lacZRi;mCherryRi*. The glial number and Dpn voxels were normalized to the control (*lacZRi;mCherryRi*) (*n* = 10, 8, 14, 12). Glial cells: Repo; NBs: Dpn. **(M, N)** Single-section (M and N) and zoomed-in images (M′ and N′) of Path-GFP expression in *elav^C155^QF>QUAS-prosRi* tumor brains, where *Tor^TED^* was overexpressed in glia using *repo-G4*, compared with *mCherryRi* at the brain surface (Yellow arrows) at 120hALH. **(O)** Quantification of the normalized (to *mCherryRi*) Path-GFP intensity at the brain surface in **(M, N)** (*n* = 10, 9). **(P)** Schematic depicting the working model. Nutrient Restriction (NR) inhibits BBB glia expansion, which consequently limits brain tumor growth. We show that NR inhibits the expression of fat body and glia-derived Ilp6, causing downregulation of the InR/PI3K pathway in BBB glia. This results in a reduction in Path expression and the downstream mTOR-S6K pathway, causing a slowdown of BBB glial cell cycle progression. Path downregulation and defects in BBB expansion consequently led to a decrease in brain AAs, such as Leu, which hinders brain tumor protein synthesis to slow down tumor growth. Data information: ALH, after larvae hatching. Brain lobes are circled with yellow dashed lines. Scale bar = 50 μm. Error bar represents SEM. In **(F)**: One-way ANOVA, (****) $P < 0.0001$; Welch's *t* test, (****) $P < 0.0001$. In **(G)**: Kruskal–Wallis test, (*) $P = 0.0391$; unpaired *t* test, (****) $P < 0.0001$. In **(I, J)**: Two-way ANOVA were used to analyze whether the effect of NR on glial number or tumor size is affected by *Rag^CA^* or *S6K^CA^* overexpression in glia (significance indicated by interaction *P* value). In (I): interaction $P < 0.0001$, interaction $P = 0.0012$. In **(J)**: interaction $P = 0.0038$, interaction $P < 0.0001$. Mean ± SEM, and statistical results including multiple comparisons are displayed in S1P and S1Q Table. In **(L)**: One-way ANOVA, (ns) $P = 0.3106$; (****) $P < 0.0001$; (***) $P = 0.0003$; (ns) $P = 0.2613$. In **(O)**: Welch ANOVA test, (ns) $P = 0.9320$. Raw data are included in S3 Data.

Our data suggests an alternative possibility that Path regulates mTOR-S6K pathway via affecting BCAA transport either directly or indirectly.

Previously, it was shown that Paneth cells act as nutrient-sensing niche cells in the mammalian gut, linking dietary cues to intestinal stem cell expansion and tissue homeostasis [71]. For instance, the NR-induced alterations in glial cell proliferative dynamics, suggest that the tumor microenvironment, much like the intestinal niche, is highly responsive to metabolic and dietary changes. These insights highlight a promising avenue for nutrient-based interventions to modulate tumor growth by leveraging the nutrient-sensing capabilities of niche cells.

In summary, our study demonstrates that targeting the Path transporter at the BBB can effectively limit brain tumor growth, while sparing normal brain tissue. Given that the BBB presents a significant challenge for drug delivery in brain cancer treatment, our findings offer a novel strategy to restrict tumor growth without the need for drugs to cross the BBB. AA transporters may serve as potential therapeutic targets given their role in cancer cell metabolism, and their involvement in the exchange of amino acids between the tumor and stroma. Further research into the specific mechanisms and vulnerabilities of these transporters could lead to the development of novel anticancer therapies that exploit the metabolic dependencies of cancer cells.

## Materials and methods

### Fly husbandry and strains

Fly strains were reared on standard food at room temperature. The standard food contains 0.42% food-grade agar, 5.25% frozen yeast, 4.38% glucose, and 4.97% Polenta. Crosses for overexpression and knockdown experiments were set up at 25 °C, and after 24 hours, the progenies were moved to 29 °C, unless otherwise stated.

The fly stocks used in this study include: *repo-GAL4* (BDSC7415), *elav-QF2^C155^* (BDSC66466), *elav-GAL4^C155^* (BDSC458), *Insc-GAL4, wor-GAL4* (from Alex Gould lab), *wrapper-GAL4* (from Owen Marshall lab), *GMR85G01-GAL4* (BDSC40436), *NP2222-GAL4* (Kyoto112830), *CG-GAL4* (BDSC7011), *Ilp2-GAL4* (from Christen Mirth lab), *Ilp6-GAL4* (KDRC103877), *tubGAL80^ts^* (BDSC7108 and 7017), *path-GFP* (from Sarah Bray lab and Jay Parrish lab) [48], *dFOXO-GFP* (generated in this study), *UAS-mCherryRNAi* (BDSC35785), *QUAS-prosRNAi* [28], *UAS-prosRNAi* (BDSC42538),

*UAS-bratRNAi* (BDSC34646), *UAS-LeuRSRNAi* (BDSC34483), *UAS-pathRNAi* (VDRC100519 and BDSC64029), *UAS-Ilp6RNAi* (BDSC33684), *UAS-myrRFP* (BDSC), *UAS-Stinger* (BDSC), *UAS-dcr2, UAS-lacZ, UAS-htl^{ACT}* (BDSC5367), *UAS-aPKC^{CAAX}* (Helena Richardson Lab), UAS-*Cdk4, UAS-CycD; +/TM6B* [72], *UAS*-S6*K^{KQ}* (BDSC6911), *UAS-Tor^{TED}* [73], *UAS-path.A* (from Susumu Hirabayashi lab and Jay Parrish lab) [50], *UAS-RagA^{CA}* (from Thomas Neufeld lab), *UAS*-S6*k^{CA}* (BDSC6914), *UAS-InR^{DN}* (BDSC8252), *UAS-InR^{CA}* (BDSC8263), *UAS-p60* [25], *UAS-myrAKT* [74], *ey-FLP1;QUAS-scribRNAi;QUAS-Ras^{V12}/CyOQS; act>CD2>QF2, UAS-RFP/TM6QS* [75], *UAS-GFP, UASp-CFP.E2f1.1-230, UASp-Venus.NLS.CycB.1–266/TM6B* (FUCCI, BDSC55114).

## Generation of dFOXO-GFP

Foxo IO-CTV-PaxCherry: vector was used to tag dFoxo with eGFP in the C-terminus. To generate the dFoxo-GFP fusion, we followed the protocol described in [76]

Using CRISPR/Cas 9, we inserted eGFP just before the stop codon in exon 10 (foxo RB). Primers and vectors sequences used for the design of IO@CTV-Pax-Cherry are available upon request.

## Dietary manipulations

For NR experiments, larvae were raised on our standard food at 25 °C and selected at the L2–L3 molting stage within a 3-hour time window based on morphological features, unless otherwise stated. Larvae were transferred to either standard food or NR, −Yeast, −Carbohydrate, +2xYeast, CDD, CDD-Leu, CDD-Ile, and CDD-Met diet (see S1R Table) in a *Drosophila* vial from either 48 or 72hALH.

## RNA sequencing

Total RNA from 5 dissected larval brains at 120hALH was pooled and extracted using Direct-zol RNA Microprep Kits (ZYMO Research, #R2061) according to the manufacturer's instructions. 5 biological replicates were prepared for each condition (Fed and NR, NR: 72-120hALH). RNA quantity and quality were assessed using Qubit and Agilent TapeStation. Two samples from NR were excluded due to low concentration or low quality. cDNA Library was prepared using the QuantSeq 3′ mRNA-Seq Library Prep Kit (Lexogen). The generated reads were aligned to the *Drosophila melanogaster* genome assembly Release 6 (Dm6) and were checked with FastQC (usegalaxy.org). Mapped reads were converted to counts using featureCounts (usegalaxy.org) and loaded to Degust (https://degust.erc.monash.edu/) for MDS plots and differential gene expression analysis using limma-voom. Differentially expressed genes were matched to genes annotated as transporters in GLAD (https://www.flyrnai.org/tools/glad/web/) [77]. Genes that are significantly downregulated (FDR ≤ 0.05, FC ≥ 1.5) were uploaded to FlyEnrichr [78,79] for KEGG pathway enrichment analysis. The volcano plot was generated using GraphPad Prism 9.0.

## Quantitative reverse transcription PCR

Total RNA from 5 dissected wild-type larval brains was pooled and extracted using Direct-zol RNA Microprep Kits (ZYMO Research, #R2061), followed by reverse transcription into cDNA using ProtoScript II First Strand cDNA Synthesis Kit (NEB, #E6560S) according to the manufacturer's instructions. 3 biological replicates were prepared for each condition (Fed and NR, NR: 63.5-96hALH). The qPCR was performed using Fast SYBR Green master mix reagent (Applied Biosystems, #4385612), on the stepOnePlus real-time PCR system (Applied Biosystems). mRNA abundance was normalized to rpl32 and calculated using the $2^{-\Delta\Delta Ct}$ method. Primers for *rpl32* [80]: Forward: CCGCTTCAAGGGACAGTATCTG; Reverse: ATCTCGCCGCAGTAAACGC. Primers for *LeuRS* (primer pair: PP36944, FlyPrimerBank, https://www.flyrnai.org/flyprimerbank): Forward: TGGCAAATTATGCAGAGTCTCG; Reverse: TGGGAAGTAGTTAAGCCAGTGTT.

## Metabolite measurements

For hemolymph sample collection, hemolymph from five tumor-bearing larvae (at 120hALH) was pooled and released into 115 µl of ultrapure water before snap-freezing in liquid nitrogen. Five biological replicates were collected for each condition (Fed and NR, NR:72-120hALH). The metabolites were extracted using 100% methanol with 20 µM of the internal standards (methionine sulfone and 2-morpholinoethanesulfonic acid), chloroform and acetonitrile [81]. Metabolites were quantified by ultra-performance liquid chromatography–tandem mass spectrometry (LC–MS-8060NX, Shimadzu) [82]. The metabolite concentrations were normalized to 2-morpholinoethanesulfonic acid, and the protein amount in the hemolymph. One sample from (NR condition) was excluded from the analysis due to variance in the internal standard detection. For brain sample collections, six brains were dissected in cold phosphate-buffered saline (PBS) and pooled into 115 µl of ultrapure water in a 1.5 ml Eppendorf tube before a brief homogenization (2 × 10 s) on ice. The tube was then snap-frozen in liquid nitrogen and stored at −80 °C before extraction and detection as described above. The concentrations of metabolites per brain were plotted.

## Immunostaining

Larval brains and SG were dissected in PBS, fixed for 20 min in 4% formaldehyde (Sigma-Aldrich, #F8775) in PBS and washed in 0.5% PBST (PBS + 0.5% TritonX-100 (Sigma-Aldrich, #T8787), 3 × 10 min). The larval fat body was fixed for 45 min and washed with 0.2% PBST. The tissues were stained with primary and secondary antibodies overnight at 4 °C. Samples were mounted in 80% glycerol in PBS or Prolong Diamond Antifade Mountant (Invitrogen, #P36961) for image acquisition. The primary antibodies used were mouse anti-Repo (1:50; DSHB, 8D12), mouse anti-Mira (1:50; gift of Alex Gould), rat anti-Mira (1:100, Abcam, #ab197788), rat anti-Dpn (1:200; Abcam, 195172), rabbit anti-pH3 (1:200, Cell Signaling, 3377), mouse anti-Elav (1:50; DSHB, 9F8A9), rabbit anti-Dcp-1-cleaved (1:100, Cell Signaling, 9578S), and mouse anti-Dacapo (1:50, DSHB). Secondary donkey antibodies conjugated to Alexa 555, 488, and Alexa 647 (Molecular Probes) were used at 1:500. Highly cross-adsorbed secondary antibodies were used to minimize cross-reactivity between mouse anti-Repo and rat anti-Dpn. DAPI and Phalloidin (both from Molecular Probes) were used at 1:1,000. Images were collected on an Olympus FV3000 confocal microscope.

## EdU labeling and analysis

EdU in vitro labeling was used to trace actively dividing tumor NBs in (S2D–S2E′ Fig). Larval brains were dissected in PBS at 120hALH and incubated in tubes with 10 µM EdU/PBS for 15 min before fixation, and primary and secondary antibody staining (for Repo and Dpn detection). EdU detection was performed using Click-iT EdU Cell Proliferation Kit for Imaging, Alexa Fluor 647 dye (Invitrogen, #C10340) following the manufacturer's instruction.

## Quantification and analysis

### Glial cell number measurements

Glial numbers were automatically counted from the 3D reconstruction of confocal Z stacks (2-µm step-size) of each larval brain lobe using a Fiji macro "DeadEasy larval Glia" [83], followed by a FIJI plugin called 3D Objects Counter.

### Size measurements

The total volume (voxels) of the tumor cell marker was measured from the 3D reconstruction of confocal Z stacks (2-µm step-size) of each sample using Volocity software to represent the tumor size. The area of the tumor brains and SGs was measured from maximum projection images, and that of each fat body cell was measured from single-section images in S1G–S1L Fig. The pupa images in (S10E and S10F Fig) were collected with a brightfield microscope and the area of the pupa was measured to represent pupa size.

## Assessment of the effect of genetic manipulations on tissue sensitivity to NR

To assess whether certain genetic manipulations can alter the effect of NR on glial and tumor growth under NR, we quantified the glial number and tumor size of each genotype under Fed and NR conditions. The Two-way ANOVA tests were used to understand whether the differences between NR and Fed are consistent between genotypes. A statistically significant interaction (interaction *P* value) means the genetic manipulation induces a significant change in the effect of NR on glial number or tumor size.

## Intensity measurements

For measurements of Path-GFP at the BBB, 8–10 lines were drawn across the brain surface glial membrane (labeled with RFP) from a single-section confocal image. Pixel values of Path-GFP and glial membrane marker RFP were generated along each line using Fiji's "Plot Profile" tool (Fig 4C). 8–10 Path-GFP pixel value of glial membrane (reflected by the peak value of RFP pixel along the line) was recorded, and used to calculate the average of Path-GFP pixel value in glial membrane of each brain lobe (Fig 4C). For Path-GFP intensity measurements in S7J–S7L Fig, sum projection confocal images were used. The intensity of the signal was measured in Fiji using the following formula: CTCF (corrected total cell fluorescence) = Integrated density − (Area of the tumor x mean fluorescence of background readings) [84]. This formula was also used to calculate BBB glia nuclear FOXO-GFP expression, and glial and fat body nuclear *Ilp6-G4>GFP* expression.

## FUCCI cell cycle analysis

FUCCI [36] was overexpressed using *GMR85GO1-G4* to assess the distribution of BBB glial cells within the cell cycle (G1: marked by ECFP; S: marked by Venus; G2/M: marked by ECFP and Venus) over time in wild-type and tumor-bearing brains. The total number of BBB glia, as well as those in each cell cycle phase, was automatically measured using a FIJI macro we previously developed [28]. The chi-squared and the Two-way ANOVA tests were used to assess the difference in the distribution of cells to each cell cycle phase. The Two-way ANOVA multiple comparisons were used to assess changes in each cell cycle phase.

## pH3 index

The NB cell cycle progression was assessed using the pH3 index, represented as the percentage of NBs in the M phase. In S1C, S1D, and S8V Figs, the number of pH3$^+$ Type I NBs and the total number of Type I NBs in the CB were counted manually in Fiji based on pH3 and Dpn or Mira staining. Type I NBs are distinguished from Type II NBs according to location and morphological features as previously described [85]. The % of pH3$^+$ Type I NBs = the number of pH3$^+$ Type I NBs/the total number of Type I NBs × 100. In Fig 3S, the ratio of NB tumor cells in M phase was assessed using the formula: The ratio of pH3$^+$ NBs = the total volume of pH3 in Dpn$^+$ cells/the total volume of Dpn in each brain lobe.

## EdU index

The cell cycle progression of tumor NBs was assessed using the EdU index, represented as the percentage of NBs that can incorporate EdU *ex vivo* within a 15-min time window. The % of EdU$^+$ NBs = the total volume (voxels) of EdU in Dpn$^+$ cells/the total volume of Dpn in each brain lobe × 100.

## The neuron ratio

The neuron ratio in each brain lobe was used to represent the dedifferentiation rate in S2D–S2F Fig). The % of Elav$^+$ cells = the total volume of Elav/the total volume of DAPI in each brain lobe × 100.

## The ratio of cells undergoing apoptosis

The ratio of cells undergoing apoptosis was assessed based on Dcp-1 staining, using the formula: The % of Dcp-1⁺ NBs = the total volume of Dcp-1 in Dpn⁺ cells/the total volume of Dpn in each brain lobe × 100. The % of Dcp-1⁺ glia = the total volume of Dcp-1 in Repo⁺ cells/the total volume of Repo in each brain lobe × 100

## Colocalisation analysis

Volocity software was used to quantify the degree of colocalization between *GMR85G01-G4>UAS-nGFP* and cell nuclear marker (either Repo or Elav in S4 A-C) in each brain lobe. Brain lobe area was selected using the ROI tool and an automatic threshold was used to exclude background noise. Pearson's Correlation was used to measure the linear correlation between the pixel intensities in two channels: −1 indicates negative correlation, 0 means no correlation and +1 suggests positive correlation. Mander's overlap coefficients (M1 and M2) were used to measure the fraction of fluorescence in one channel that overlaps with signal in the other, which indicates the co-occurrence of two channels. M1: Proportion of channel 1 (green) overlapping with channel 2 (magenta). M2: Proportion of channel 2 overlapping with channel 1.

## Statistical analysis

Shapiro–Wilk test was used to assess data normality. Two-tailed unpaired Student *t* tests and One-way ANOVA tests were used to assess the difference between two groups and three or more independent groups, respectively. The non-parametric Mann–Whitney *U* test and Kruskal–Wallis test were used for not normally distributed data. The Welch's correction and Brown–Forsythe correction were used for the data of unequal variance. The chi-squared test was used to test whether two categorical variables are related to each other. The Two-way ANOVA tests were used to understand the interaction between genotypes and nutritional status in regulating glial number and tumor size, as well as the effect of developmental timing on the percentage of BBB glia in each cell cycle phase.

## Supporting information

**S1 Fig. Characterization of the effect of nutrient restriction (NR) on the growth of peripheral tissues and neuroblast (NB) tumors (related to Fig 1). (A, B)** Single-section images of wild-type brain lobes (ventral side) stained with the Miranda (Mira) and the mitotic marker pH3 under Fed and NR conditions. CBs are circled by yellow dashed lines. NR: 63.5-96hALH; Dissection: 96hALH. Genotype: *repo-G4>UAS-RFP*. **(C, D)** Quantification of type I NB number (based on Mira staining) and the percentage of type I NBs undergoing mitosis (pH3⁺Mira⁺) in the CB in (A, B) (*n* = 8, 8). **(E)** Schematic representation of the Fed/NR regime. **(F)** Table depicting the consequence of NR starting from 48, 63.5, 68, 72, and 96hALH on pupariation of *elav^C155^QF>QprosRNAi* tumor-bearing animals. **(G–I)** NR from 68, 72, and 96hALH significantly reduced the size of fat body cells, salivary glands, and brain tumors (circled by yellow dashed lines). (G, H): single-section images of fat body and maximum projection images of salivary glands, marked by DAPI and Phalloidin at NR starting time point, Fed$_{120hALH}$ and NR$_{120hALH}$. (I): maximum projection images of brain tumors, marked by DAPI at NR starting time point, Fed$_{120hALH}$ and NR$_{120hALH}$. Scale bar = 100 μm for (G) and (H). **(J–L)** Quantifications of the area of a single fat body cell circled in (G), the salivary gland circled in (H) and the whole brain circled in (I). In (J): $n_{68hALH}$ = 105, 60, 75; $n_{72hALH}$ = 79, 52, 68; $n_{96hALH}$ = 74, 62, 58. In (K): $n_{68hALH}$ = 8, 13, 12; $n_{72hALH}$ = 11, 23, 19; $n_{96hALH}$ = 11, 5, 9. In (L): $n_{68hALH}$ = 5, 9, 8; $n_{72hALH}$ = 13, 12, 12; $n_{96hALH}$ = 8, 5, 5. Genotype: *elav^C155^QF>QprosRNAi*. Data information: ALH = after larvae hatching. Scale bar = 50 μm unless otherwise stated. The error bar represents SEM. In (C): unpaired *t* test, (ns) *P* = 0.8060. In (D): unpaired *t* test, (ns) *P* = 0.7211. In (J): Kruskal–Wallis H Test, (****) *P* < 0.0001; Kruskal–Wallis H Test, (****) *P* < 0.0001; Kruskal–Wallis H Test, (****) *P* < 0.0001. In (K): Kruskal–Wallis H Test, (****) *P* < 0.0001; Kruskal–Wallis H Test, (***) *P* = 0.0002; Kruskal–Wallis H Test, (**) *P* = 0.0011. In (L): (***) *P* = 0.0006; Welch's ANOVA, (****) *P* < 0.0001; Welch's ANOVA, (**) *P* = 0.0020. Raw data are included in S3 Data.
(TIFF)

**S2 Fig. The effects of nutrient restriction (NR) on the proliferation, differentiation and apoptosis of Pros-loss-of-function neuroblast (NB) tumors (related to Fig 1). (A–B′)** Single-section images of *elav^C155^QF>QUAS-prosRi* tumor brains labeled with EdU (yellow arrows) and Dpn under Fed and NR conditions (the same dataset from Fig 1J–1L). A and B: Dpn and EdU; A′ and B′: EdU. **(C)** Quantification of the percentage of NB tumors (Dpn⁺, green) that are EdU⁺ in each brain lobe of (A, B) ($n = 14, 10$). **(D–E′)** Single section images of *elav^C155^QF>QUAS-prosRi* tumor brains stained with DAPI and the neuronal marker Elav under Fed and NR conditions. **(F)** Quantification of the percentage of neurons among all brain cells in each brain lobe of (D–E′) ($n = 12, 10$). **(G–H′)** Single section images of *elav^C155^QF>QUAS-prosRi* tumor brains stained with Dpn and the cell death marker Dcp-1 (yellow arrows) under Fed and NR conditions. (G′) and (H′) are zoomed-in images of (G) and (H). **(I)** Quantification of the percentage of NBs (Dpn⁺) undergoing apoptosis (Dcp-1⁺) of each circled brain lobe in (G, H) ($n = 8, 6$). Data information: ALH = after larvae hatching. NR: 72-120hALH; Dissection: 120hALH unless otherwise stated. Brain lobes are circled with yellow dashed lines. Scale bar = 50 μm. Error bar represents SEM. In (C): unpaired *t* test, (*) $P = 0.0128$. In (F): unpaired *t* test, (*) $P = 0.0247$. In (I): Welch's *t* test, (ns) $P = 0.5374$. Raw data are included in S3 Data.
(TIFF)

**S3 Fig. The effect of nutrient restriction (NR) on the growth of other brain tumors and characterization of the *GMR85G01-G4* driver expression pattern (related to Fig 2). (A, B)** Maximum projection images of *insc-G4,wor-G4>aPKC-CAAX* tumor brains stained with Repo under Fed and NR conditions. **(C)** Quantification of normalized (to Fed) glial number of circled brain lobes in (A, B) ($n = 13, 13$). **(D, E)** Maximum projection images of *elav^C155^-G4>bratRi* tumor brains stained with Repo under Fed and NR conditions. **(F)** Quantification of normalized (to Fed) glial number of circled brain lobes in (D, E) ($n = 16, 20$). **(G, H)** Single-section images of *elav^C155^QF>QUAS-prosRi* tumor brain lobes, where SPG are marked with *moody-G4>UAS-Stinger*, under fed and NR conditions. NR:72-96hALH. Dissection: 96hALH. **(I)** Quantifications of SPG numbers (normalized to fed) in (G, H) ($n = 8, 10$). **(J)** Quantification of SPG nuclear size (normalized to fed) in (G, H) ($n = 8, 8$). **(K, L)** Maximum projection images of *elav^C155^QF>QUAS-prosRi* tumor brain lobes, where CG are marked with *wrapper-G4>UAS-RFPnls*, under fed and NR conditions. NR:72-96hALH. Dissection: 96hALH. **(M)** Quantification of CG number (normalized to fed) in (K, L) ($n = 8, 5$). Data information: ALH = after larvae hatching. NR: 72-120hALH; Dissection: 120hALH unless otherwise stated. Brain lobes are circled with yellow dashed lines. Scale bar = 50 μm. Scale bar = 20 μm in K′, L′, and M′ Error bar represents SEM. In (C): unpaired *t* test, (****) $P < 0.0001$. In (F): unpaired *t* test, (****) $P < 0.0001$ In (I): Welch's *t* test, (**) $P = 0.0054$. In (J): Welch's *t* test, (ns) $P = 0.069$. In (M): unpaired *t* test, (**) $P = 0.0021$. Raw data are included in S3 Data.
(TIFF)

**S4 Fig. The effect of nutrient restriction (NR) on other glial cell types and glial cell death. (A–A‴)**: Deep-section image of late L3 wildtype larval brain lobe, where *GMR85G01-G4>UAS-Stinger* showed no colocalization with neurons (marked by Elav in magenta). A′–A‴ are zoomed-in images of A. **(B–B‴)**: Surface-section image of late L3 wildtype larval brain lobe, where *GMR85G01-G4>UAS-Stinger* colocalised with some glial cells (marked by Repo). These GFP⁺ glial cells include PG (yellow arrows) and SPG (with enlarged nuclei, caused by endoreplication, blue arrows). And those GFP⁻ glial cells are surface-associated CG. B′–B‴ are zoomed-in images of B. **(C–C‴)**: Mid-section image of late L3 wildtype larval brain lobe, where *GMR85G01-G4>UAS-Stinger* did not colocalise with the glial cell marker Repo. These glial cells are CG (gray arrows) based on their location. **(D)**: Quantifications of Pearson's correlation coefficient (PCC) and Mander's Overlap Coefficient (as explained in Materials and methods) ($n = 8, 5$). **(E–H)**: *GMR85G01-G4* is expressed in larval gut, salivary gland (SG) and wing discs (WD), but not fat body (FB), shown by inducing *UAS-Stinger* in the cell nucleus ($n = 3, 3, 3, 3$). Scale bar: 200 μm in (E) and 100 μm in (F–H). **(I, J)** Time-course single-section images of wild-type (I–I‴) and *elav^C155^QF>QUAS-prosRi* tumor brains (J–J‴) with FUCCI overexpressed in BBB glia using *GMR85G01-G4* (G1-phase cells: red; S-phase cells: green; and G2/M-phase cells: yellow). Time points: 48, 72, 96hALH and 24hAPF in wild-type

larvae; and 48, 72, 96, and 120hALH in tumor-bearing larvae (pupate between 120-144hALH). **(K, L)** Single section images of *elav^C155^QF>QUAS-prosRi* tumor brains stained with Repo and the cell death marker Dcp-1 (yellow arrows) under Fed and NR conditions. (K′) and (L′) are zoomed-in images of K and L. **(M)** Quantification of the percentage of glial cells (Repo⁺) undergoing cell death (Dcp-1⁺ Repo⁺) among all glia in each brain lobe in (K, L) ($n = 8, 6$). Data information: ALH, after larvae hatching; APF, after pupa formation. NR: 72-120hALH; Dissection: 120hALH unless otherwise stated. Brain lobes are circled with yellow dashed lines. Scale bar = 50 μm. Error bar represents SEM. In (D): For GFP and Elav colocalization, Pearson's correlation: $-0.099 \pm 0.024$, M1 coefficient: $0.05 \pm 0.01$, suggesting low fraction of GFP overlaps with Elav staining. M2 coefficient: $0.04 \pm 0.01$, meaning low fraction of Elav overlaps with GFP. For GFP and Repo colocalization, Pearson's correlation: $0.5728 \pm 0.038$, M1 coefficient: $0.94 \pm 0.04$, suggesting majority of GFP signals overlaps with Repo staining. M2 coefficient: $0.20 \pm 0.02$, suggesting a proportion (20%) of Repo overlaps with GFP. In (M): Mann–Whitney test, (ns) $P = 0.6620$. Raw data are included in S3 Data. (TIFF)

**S5 Fig. The effect of nutrient restriction (NR) on hemolymph and tumor brain amino acid content (related to Fig 3). (A)** Quantification of Fed versus NR hemolymph AA concentrations of *elav^C155^QF>QUAS-prosRi* tumor-bearing animals ($n = 5, 4$). **(B)** Quantifications of the normalized (to Fed) brain AA concentration in *elav^C155^QF>QUAS-prosRi* tumor-bearing animals under Fed and NR ($n = 5, 5$). Data information: NR: 72-120hALH; Dissection: 120hALH unless otherwise stated. Error bar represents SEM. In (A): unpaired $t$ test, (ns) $P = 0.8631$; (**) $P = 0.0091$; (***) $P = 0.0001$; (***) $P = 0.0002$; (***) $P = 0.0005$; (****) $P < 0.0001$; (**) $P = 0.0018$; (***) $P = 0.0004$; (***) $P = 0.0001$; (*) $P = 0.0322$; (****) $P < 0.0001$; (***) $P = 0.0002$; (ns) $P = 0.1710$; (***) $P = 0.0002$; (**) $P = 0.0016$; (ns) $P = 0.9419$; (ns) $P = 0.2580$; (****) $P < 0.0001$; (****) $P < 0.0001$. In (B): Lys: unpaired $t$ test, (*) $P = 0.0402$; Met: unpaired $t$ test, (**) $P = 0.0048$; Phe: Mann–Whitney test, (*) $P = 0.0317$; Val: unpaired $t$ test, (****) $P < 0.0001$; Ile: unpaired $t$ test, (**) $P = 0.0059$; Thr: Welch's $t$ test, (**) $P = 0.0015$; Arg: unpaired $t$ test, (*) $P = 0.0138$; Gly: unpaired $t$ test, (*) $P = 0.0334$; Glu: Mann–Whitney test, (**) $P = 0.0079$; Ala: unpaired $t$ test, (****) $P < 0.0001$; Asn: unpaired $t$ test, (***) $P = 0.0001$; Asp: unpaired $t$ test, (***) $P = 0.0003$; Pro: Mann–Whitney test, (**) $P = 0.0079$; Ser: unpaired $t$ test, (***) $P = 0.0002$; Gln: unpaired $t$ test, (****) $P < 0.0001$; Tyr: Welch's $t$ test, (*) $P = 0.0453$. Raw data are included in S3 Data. (TIFF)

**S6 Fig. Nutrient restriction (NR) downregulates BCAA tRNA synthetases specifically in tumor brains, but not in wild-type brains (related to Fig 3). (A)** Multidimensional Scaling (MDS) plot, showing gene expression profiles of *elav^C155^QF>QUAS-prosRi* brain tumor samples are separated by nutrient conditions (Fed versus NR). NR: 72-120hALH. **(B)** The variance table displaying how much each dimension in the MDS plot accounts for the total variance in the data. **(C)** KEGG pathway analysis showing top 10 downregulated pathways, ranked by the FlyEnrichr combined score. **(D)** Plot of transcript abundance alteration of *LeuRS*, *IleRS*, *ValRS*, *Bcat*, *Bckdha*, *Bckdhb* and *Dbct* in the tumor brains under NR versus Fed (from RNA seq data, $n = 5, 3$). **(E, F)** Maximum projection images of wild-type brains, where *mCherryRi* or *LeuRSRi* was overexpressed in NBs using *elav^C155^-G4,GAL80^ts^*. Larvae were raised at 18 °C for 4 days before being moved to 29 °C for transgene activation (2 days). Brains are stained with DAPI. **(G)** Quantifications of the normalized (to *mCherryRi*) DAPI voxels of each brain in (E and F) ($n = 6, 4$). **(H)** Quantification of mRNA level of *LeuRS* in wild-type brains under NR versus fed (by RT-qPCR, $n = 3,3$). NR: 63.5-96hALH. Data information: NR: 72–120 hALH; Dissection: 120hALH unless otherwise stated. Error bar represents SEM. In (D): (*) FDR = 0.024; (*) FDR = 0.043; (ns) FDR > 0.05. In (G): unpaired $t$ test, (****) $P < 0.0001$. In (H): unpaired $t$ test, (ns) $P = 0.3666$. Raw data are included in S3 Data. (TIFF)

**S7 Fig. Characterization of the effect of nutrient restriction (NR) on Path expression in other brain cell types and other tumors (related to Fig 4). (A–B″)** Single-section (A and B) and zoomed-in images (A′–B″) of Path-GFP expression in control versus *elav^C155^QF>QUAS-prosRi* tumor brains at 96hALH. Glia at the brain surface are marked

with *repo-G4>mRFP* and distinguished from the other glial types based on position (yellow arrows). Scale bar = 20 μm in (A′–B″). **(C)** Quantification of the normalized (to control) Path-GFP intensity at the BBB in (A′ and B′) (*n* = 10, 8). **(D–E)** Single-section (D and E) and zoomed-in images (D′–E″) of Path-GFP expression in wild-type brains under Fed and NR. Glial membrane at the brain surface is marked with *repo-G4>mRFP* (yellow arrows). NR: 65 -96hALH; Dissection: 96hALH. Scale bar = 20 μm in (D′–E″). **(F)** Quantification of the normalized (to Fed) Path-GFP intensity at the BBB in (D and E) (*n* = 8, 8). **(G–H″)** Single-section images of Path-GFP expression in NBs (Mira, blue arrows) and CG (*repo-G4>RFP*, magenta arrows, distinguished from other glia based on location) in the *elav^C155^QF>QUAS-prosRi* tumor brains under Fed and NR. Scale bar = 20 μm. **(I)** Quantification of the normalized (to Fed) Path-GFP intensity of NBs and CG in (G–H″) (*n* = 63, 45, 60, 35). **(J, K)** Maximum projection images of Path-GFP expression in *ey-FLP, act-G4>Ras^V12^, scribRi* eye disc tumors, marked by *UAS-RFP* under Fed and NR. **(L)** Quantification of Path-GFP intensity (CTCF, described in [Materials and methods]) of circled tumor normalized to the tumor area in (J, K) (*n* = 6, 6). (J–L) are from the same experiment as [Fig 1W]–[1Y]. Data information: ALH = hours after larvae hatching. NR: 72-120hALH; Dissection: 120hALH unless otherwise stated. Brain lobes are circled with yellow dashed lines. Scale bar = 50 μm. Error bar represents SEM. In (C): unpaired *t* test, (ns) *P* = 0.6740. In (F): unpaired *t* test, (*) *P* = 0.0144. In (I): Mann–Whitney test, (*) *P* = 0.0240; Mann–Whitney test, (****) *P* < 0.0001. In (L): Mann–Whitney test, (ns) *P* = 0.1320. Raw data are included in [S3 Data]. (TIFF)

**S8 Fig. Characterization of the role of Path in tumor glia and the wild-type glia (related to [Fig 4]). (A, B)** Single section images of *elav^C155^QF>QUAS-prosRi* tumor brains, where *mCherryRi* and *pathRi^KK^* were overexpressed in glia using *repo-G4*. Glia: Repo; NBs: Dpn. **(C, D)** Quantifications of the normalized (to *mCherryRi*) glial number (C) and Dpn voxels (D) of each circled brain lobe in (A, B) (*n* = 10, 14). **(E–F)** Single-section (E and F) and zoomed-in images (E′–F‴) of Path-GFP expression (yellow arrows) in *elav^C155^QF>QUAS-prosRi* tumor brains, where *pathRi^KK^* was overexpressed in BBB glia using *GMR85G01-G4*, compared with *control* (CyOYFP siblings) at 96hALH. Scale bar = 20 μm in (E′–F‴). **(G)** Quantification of the normalized (to *control*) Path-GFP intensity at the BBB in (E–F) (*n* = 4, 7). **(H, I)** Single section images of *elav^C155^QF>QUAS-prosRi* tumor brains with *pathRi^KK^* (I) overexpressed specifically in CG using *NP2222-G4*, compared to *mCherryRi* (H) at 120hALH. Glia: Repo; NBs: Dpn. **(J, K)** Quantifications of the normalized (to *mCherryRi*) glial number (J) and Dpn voxels (K) of each circled brain lobe in (H, I) (*n* = 14, 11). **(L, M)** Single section images of *elav^C155^QF>QUAS-prosRi* tumor brains with an independent *pathRi* co-overexpressed with *dcr2* (M) in glia using *repo-G4*, compared to *dcr2, lacZRi* (L) at 120hALH. Glia: Repo; NBs: Dpn. **(N, O)** Quantifications of the normalized (to *dcr2, lacZRi*) glial number (N) and Dpn voxels (O) of each circled brain lobe in (L, M) (*n* = 10, 11). **(P, Q)** Maximum projection images of wild-type brain lobes with *pathRi^KK^* (Q) overexpressed in glia using *repo-G4*, compared to *mCherryRi* (P) at 96hALH. Glia: Repo. **(R)** Quantifications of the normalized (to *mCherryRi*) glial number of each circled brain lobe in (P, Q) (*n* = 12, 12). **(S–T′)** Wild-type brain lobes where *pathRi^KK^* was overexpressed in glia using *repo-G4* under Fed and NR. Glia: Repo (S and T, maximum projection), Dpn and pH3 in (S′ and T′, single sections). CBs are circled by yellow dashed lines. NR: 65-89hALH; Dissection: 89hALH. **(U)** Quantification of the normalized (to Fed) glial number of each circled brain lobes in (S and T) (*n* = 14, 14). **(V)** Quantification of the percentage of NBs undergoing mitosis (pH3⁺Dpn⁺) in the CB of (S′ and T′) (*n* = 14, 14). Data information: ALH = hours after larvae hatching. Brain lobes are circled with yellow dashed lines. Scale bar = 50 μm. Error bar represents SEM. In (C): One-way ANOVA, (****) *P* < 0.0001. In (D): Kruskal–Wallis test, (****) *P* < 0.0001. In (G): One-way ANOVA, (**) *P* = 0.0015. In (J): unpaired *t* test, (ns) *P* = 0.1049. In (K): unpaired *t* test, (ns) *P* = 0.6000. In (N): Mann–Whitney test, (***) *P* = 0.0001. In (O): Welch's *t* test, (****) *P* < 0.0001. In (R): unpaired *t* test, (ns) *P* = 0.1245. In (U): unpaired *t* test, (****) *P* < 0.0001. In (V): unpaired *t* test, (****) *P* < 0.0001. Raw data are included in [S3 Data]. (TIFF)

**S9 Fig. The effect of BBB Path overexpression on brain tumor AA content. (A–R)** Quantifications of the normalized (to *lacZ^OE^_*Fed) brain AA concentration in *elav^C155^QF>QUAS-prosRi* tumor-bearing animals, where *lacZ* or *path* is overexpressed

using *GMR85G01-G4* under Fed and -Yeast conditions (*n* = 10, 10, 4, 7). Data information: Two-way ANOVA were used to analyze whether the effect of yeast dropout on brain AA concentration is restored upon Path overexpression in the BBB. Statistical results including multiple comparisons are displayed in S1M Table. Raw data are included in S3 Data. (TIFF)

**S10 Fig. The glial PI3K pathway is not regulated by systemic Ilps (related to Fig 6). (A)** Quantification of Path-GFP expression at the surface glia of the *elav^C155^QF>QUAS-prosRi* tumor brains with *InR^DN^*, *p60* or *InR^CA^* overexpressed in glia using *repo-G4* at 120hALH (*n* = 14, 13, 12, 8). **(B, C)** surface-section images and zoomed-in images of Foxo-GFP expression (Yellow arrows) in *elav^C155^QF>QUAS-prosRi* tumor brains, where *lacZRi* or *pathRi^KK^* was overexpressed using *repo-G4*. Glia: Repo. Dissection: 96hALH. Scale bar = 50 μm in B and C. Scale bar = 20 μm in zoomed-in images (1–4). **(D)** Quantification of glial nuclear Foxo-GFP intensity in (B, C) (*n* = 6, 12). **(E, F)** Pupa, salivary glands (SG, stained with Phalloidin and DAPI) and brains (stained with Repo and Dpn) in *elav^C155^QF>QUAS-prosRi* tumor-bearing animals, where *p60* was overexpressed in insulin-producing cells (IPCs) using *Ilp2-G4*, compared with *mCherryRi* at 120hALH. **(E′)** and **(F′)**: maximum projection images; **(E″)** and **(F″)**: single section images; Scale bar = 1 mm in (E and F). **(G)** Quantification of the normalized (to *mCherryRi*) pupal size (area) in (E and F) (*n* = 5, 4). **(H)** Quantification of the normalized (to *mCherryRi*) SG size (area) in (E′ and F′) (*n* = 8, 7). **(I, J)** Quantifications of the normalized (to *mCherryRi*) glial number (I) and Dpn voxels (J) of each circled brain lobe in (E″ and F″) (*n* = 14, 10). **(K)** Quantifications of the GFP intensity (CTCF) in BBB glial nucleus in wild-type and *elav^C155^QF>QUAS-prosRi* tumor brains under Fed and NR conditions. Wild-type and tumor-bearing animals were starved after CW (65hALH and 68hALH, respectively) (*n* = 120, 120, 60, 100 cells from 12, 12, 6, 10 brain lobes). Data information: ALH = after larvae hatching. Brain lobes are circled with yellow dashed lines. Scale bar = 50 μm. Error bar represents SEM. In (A): One-way ANOVA, (****) *P* < 0.0001, (**) *P* = 0.0073, (****) *P* < 0.0001. In (D): unpaired *t* test, (***) *P* = 0.0001. In (G): unpaired *t* test, (**) *P* = 0.0098. In (H): unpaired *t* test, (****) *P* < 0.0001. In (I): unpaired *t* test, (****) *P* < 0.0001. In (J): Welch's *t* test, (***) *P* = 0.0001. In (K): Two-way ANOVA was used to analyze whether the effect of NR on BBB glial Ilp6 expression is different in the tumor brains compared with wild-type brains (significance indicated by interaction *P* value). Interaction *P* = 0.1876. Multiple comparisons: WT fed versus NR: (ns) *P* = 0.2488; Tumor fed versus NR: (**) *P* = 0.0095; WT fed versus Tumor fed: (****) *P* < 0.0001; WT NR versus Tumor NR: (****) *P* < 0.0001. Raw data are included in S3 Data. (TIFF)

**S1 Data. Excel spreadsheet showing genes significantly upregulated and downregulated in tumor brains under NR compared to Fed (False Discovery Rate (FDR) ≤0.05, Fold Change (FC) ≥ 1.5).** (XLSX)

**S2 Data. Excel spreadsheet showing the expression of annotated transporters (using GLAD) in tumor brains under fed and NR conditions (False discovery rate, FDR < 0.05; Fold change (FC) >1.5).** (XLSX)

**S3 Data. Excel spreadsheet showing all the raw numerical values displayed in each figure.** (XLSX)

**S1 Table. This sheet includes Statistical data and the food recipe mentioned in Materials and methods.** (DOCX)

## Acknowledgments

We are grateful to Sarah Bray, Jay Parrish, Owen Marshall, Christen Mirth, Helena Richardson, Susumu Hirabayashi, Christian F. Lehner, Thomas Neufeld, Alex Gould and Jean-Paul Vincent for the generous sharing of antibodies and fly stocks. We would like to thank Rina Okada from the Obata lab for metabolomics expertise, Andrew Cox and Owen

Marshall for critical reading of the manuscript. We also thank the Bloomington *Drosophila* Stock Centre (BDSC), Vienna *Drosophila* Resource Centre (VDRC), Kyoto *Drosophila* Stock Centre (KDRC) and Developmental Studies Hybridoma Bank for fly stocks and antibodies. We would also like to thank OZDros for *Drosophila* quarantine, Peter MacCallum Center for Advanced Histology and Microscopy for microscopy assistance, and the molecular genomics core for sequencing support.

## Author contributions

**Conceptualization:** Qian Dong, Edel Alvarez-Ochoa, Phuong-Khanh Nguyen, Paul Orih, Natasha Fahey-Lozano, Louise Y. Cheng.

**Data curation:** Phuong-Khanh Nguyen, Paul Orih, Natasha Fahey-Lozano.

**Formal analysis:** Qian Dong, Edel Alvarez-Ochoa, Phuong-Khanh Nguyen, Paul Orih, Natasha Fahey-Lozano, Hina Kosakamoto.

**Funding acquisition:** Louise Y. Cheng.

**Investigation:** Qian Dong, Edel Alvarez-Ochoa, Natasha Fahey-Lozano, Louise Y. Cheng.

**Methodology:** Qian Dong, Edel Alvarez-Ochoa, Hina Kosakamoto, Cyrille Alexandre.

**Project administration:** Qian Dong, Louise Y. Cheng.

**Resources:** Fumiaki Obata, Louise Y. Cheng.

**Supervision:** Fumiaki Obata, Louise Y. Cheng.

**Validation:** Qian Dong, Edel Alvarez-Ochoa, Hina Kosakamoto, Louise Y. Cheng.

**Writing – original draft:** Qian Dong, Louise Y. Cheng.

**Writing – review & editing:** Qian Dong, Edel Alvarez-Ochoa, Louise Y. Cheng.

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
