## [Editor Report · Decision Letter 0]

19 Feb 2025

Dear Dr Cheng,

Thank you for submitting your manuscript entitled "The blood-brain barrier regulates brain tumour growth specifically via the SCL36 amino acid transporter Pathetic in Drosophila" for consideration as a Research Article by PLOS Biology.

Your manuscript has now been evaluated by the PLOS Biology editorial staff as well as by an academic editor with relevant expertise and I am writing to let you know that we would like to send your submission out for external peer review.

Once your full submission is complete, your paper will undergo a series of checks in preparation for peer review. After your manuscript has passed the checks it will be sent out for review. To provide the metadata for your submission, please Login to Editorial Manager (https://www.editorialmanager.com/pbiology) within two working days, i.e. by Feb 21 2025 11:59PM.

Kind regards,

Ines

--

Ines Alvarez-Garcia, PhD

Senior Editor

PLOS Biology

---

## [Decision Letter · Decision Letter 1]

18 Apr 2025

Dear Dr Cheng,

Thank you for your patience while your manuscript entitled "The blood-brain barrier regulates brain tumour growth specifically via the SCL36 amino acid transporter Pathetic in Drosophila" was peer-reviewed at PLOS Biology. Your manuscript has been evaluated by the PLOS Biology editors, an Academic Editor with relevant expertise, and by two independent reviewers.

The reviews are attached below. As you will see, the reviewers find the conclusions interesting and novel, but they also raise several issues that would need to be addressed before we can consider the manuscript for publication. Reviewer 1 mentions that the evidence suggesting that Path levels correlate with brain BCAA levels should be strengthened, and that the link between the PI3K/AKT and Ilp6 signalling pathways and the functional regulation of BCAA levels or transport remains unclear, along with whether or not the upstream pathways act via Path and BCAA availability. The reviewer suggests several experiments to confirm the model and other conclusions. Reviewer 2 raises some issues regarding the Path::GFP stainings and thinks that western blots should be used to demonstrate the expression changes, that important controls are missing, and also that the claim that nutrient restriction regulates PI3K signalling in perineurial glia is not supported by the data.

After discussing the reviews with the Academic Editor, we do think you will need to address the following points as essential for publication:

- The evidence suggesting that Path levels correlate with brain BCAA levels could be strengthened. In Figure 4, you show that glial-specific knockdown of path under nutrient restriction reduces tumor growth and brain amino acid levels—particularly BCAAs—without affecting known Path substrates, suggesting a broader role for Path in regulating amino acid availability. While path knockdown (KD) was assessed for its effect on brain amino acid concentrations, no equivalent analysis was performed in the context of Path overexpression, even though it is shown that Path overexpression in PG cells partially rescues tumor growth under NR conditions (Fig. 4I-J). It remains unclear whether the observed phenotypic rescue correlates with restored levels of BCAAs.

- The Fly-FUCCI system driven by the PG-specific GMR85G01-GAL4 driver is used to monitor cell cycle dynamics in perineural glia (PG) under fed and NR conditions. You report a significant increase in G1-phase cells and a corresponding decrease in S-phase cells under NR, suggesting a slowdown or arrest of the cell cycle. You also state that this change is not due to increased cell death, thereby implying true cell cycle arrest. However, the FUCCI system alone is insufficient to conclusively demonstrate cell cycle arrest. FUCCI provides a proxy for cell cycle phase distribution but does not differentiate between transient delay and true checkpoint-mediated arrest. To substantiate the claim of cell cycle arrest, you should perform additional experiments such as analyzing the expression of phase-specific markers (e.g., Cyclins) and examining well-established cell cycle arrest or checkpoint markers, including p21, p27, phosphorylated Chk1/Chk2, or γH2AX in the case of DNA damage-induced arrest.

- For statistical analysis, it seems that fed was compared to NF; however, within group analysis (fed control compared to KD or OE) is not shown for some of the plots. This can result in misinterpretation of the results (e.g. Fig 2S-T). Based on the data, it seems that CDk4OE; CycDOE have increased glial numbers independent of fed/NR.

- It is not entirely clear why the NR timelines are different between figures and even between WT and tumor (e.g. Fig 2 & 4 NR was from 72-96h while in Fig 6. The WT was NR from 65h to 92h while tumor was from 68h to 92h). Fig 3, the =Y or -GP was fed for much longer (48-120h). It is difficult to make comparisons among the groups this way.

In addition, while you don't need to do everything, you should try something suggested here:

- To strengthen the mechanistic model, you should perform Path overexpression and/or BCAA supplementation experiments in the context of PI3K or Ilp6 inhibition to determine whether BCAAs can rescue the glial and tumor phenotypes. Conversely, it would be important to test whether the effects of these pathways are diminished or abolished by path knockdown, thereby establishing Path as a functional mediator. Additionally, measuring BCAA levels under these genetic conditions would help directly link pathway activity, Path expression, and amino acid availability. Without such functional validation, the proposed model connecting upstream signals to tumor growth via Path-BCAA regulation remains speculative.

We appreciate that these requests represent a great deal of extra work, and we are willing to relax our standard revision time to allow you 6 months to revise your study. Please email us (plosbiology@plos.org) if you have any questions or concerns, or envision needing a (short) extension.

**IMPORTANT - SUBMITTING YOUR REVISION**

3. Resubmission Checklist

a) *PLOS Data Policy*

b) *Published Peer Review*

Sincerely,

Ines

--

Ines Alvarez-Garcia, PhD

Senior Editor

PLOS Biology

Reviewers' comments

Rev. 1:

Dong et al. present a study identifying the amino acid transporter Pathetic (Path) at the blood-brain barrier (BBB) as a key regulator of brain tumor sensitivity to nutrient restriction (NR), revealing the molecular role of the perineurial glial BBB cells in controlling tumor growth. The authors provide strong evidence that neuroblasts tumors lose the ability to buffer against NR through the path transporter. However, the manuscript lacks sufficient experimental evidence to fully support its mechanistic claims that the NR buffering capacity is due to brain BCAA levels. Several points are listed below that would strengthen the conclusions if addressed:

The evidence suggesting that Path levels correlate with brain BCAA levels could be strengthened. In Figure 4, the authors show that glial-specific knockdown of path under nutrient restriction reduces tumor growth and brain amino acid levels—particularly BCAAs—without affecting known Path substrates, suggesting a broader role for Path in regulating amino acid availability. While path knockdown (KD) was assessed for its effect on brain amino acid concentrations, no equivalent analysis was performed in the context of Path overexpression, even though the authors show (Fig. 4I-J) that Path overexpression in PG cells partially rescues tumor growth under NR conditions. It remains unclear whether the observed phenotypic rescue correlates with restored levels of BCAAs.

From Figure 5-7, the authors explore the upstream regulation of Path through the PI3K/AKT and Ilp6 signaling pathways, suggesting a mechanistic framework for how nutrient restriction downregulates Path expression. However, the link between these signaling mechanisms and the functional regulation of BCAA levels or transport remains insufficiently tested.

While Figure 7 shows that path knockdown impairs tumor growth and that its expression is necessary for tumor recovery, it remains unclear whether the upstream pathways act specifically through Path and BCAA availability.

To strengthen the mechanistic model, I suggest performing Path overexpression and/or BCAA supplementation experiments in the context of PI3K or Ilp6 inhibition to determine whether BCAAs can rescue the glial and tumor phenotypes. Conversely, it would be important to test whether the effects of these pathways are diminished or abolished by path knockdown, thereby establishing Path as a functional mediator. Additionally, measuring BCAA levels under these genetic conditions would help directly link pathway activity, Path expression, and amino acid availability. Without such functional validation, the proposed model connecting upstream signals to tumor growth via Path-BCAA regulation remains speculative.

The authors assert that Path is a BCAA transporter Lines 460-461 "these data suggest that PG Path, rather than transporting its known substrates, likely regulates glial and tumour growth by controlling the transport of other AAs, including BCAAs.", but the evidence for this is weak. If the authors are shifting the known paradigm, they should provide evidence for BCAA transport through Path.

The authors use the Fly-FUCCI system driven by the PG-specific GMR85G01-GAL4 driver to monitor cell cycle dynamics in perineural glia (PG) under fed and NR conditions. They report a significant increase in G1-phase cells and a corresponding decrease in S-phase cells under NR, suggesting a slowdown or arrest of the cell cycle. The authors also state that this change is not due to increased cell death, thereby implying true cell cycle arrest. However, the FUCCI system alone is insufficient to conclusively demonstrate cell cycle arrest. FUCCI provides a proxy for cell cycle phase distribution but does not differentiate between transient delay and true checkpoint-mediated arrest. To substantiate the claim of cell cycle arrest, I suggest performing additional experiments such as analyzing the expression of phase-specific markers (e.g., Cyclins) and examining well-established cell cycle arrest or checkpoint markers, including p21, p27, phosphorylated Chk1/Chk2, or γH2AX in the case of DNA damage-induced arrest.

For statistical analysis, it seems that fed was compared to NF; however, within group analysis (fed control compared to KD or OE) is not shown for some of the plots. This can result in misinterpretation of the results (e.g. Fig 2S-T). Based on the data, it seems that CDk4OE; CycDOE have increased glial numbers independent of fed/NR.

It is not entirely clear why the NR timelines are different between figures and even between WT and tumor (e.g. Fig 2 & 4 NR was from 72-96h while in Fig 6. The WT was NR from 65h to 92h while tumor was from 68h to 92h). Fig 3, the =Y or -GP was fed for much longer (48-120h). It is difficult to make comparisons among the groups this way.

Minor comments:

1. In Figure 4M, the authors report that BCAA levels are reduced following path knockdown, supporting the idea that Path modulates amino acid availability. While this trend is generally supported by the data, it should be noted that isoleucine (Ile) does not show a statistically significant decrease, which somewhat weakens the claim of a consistent BCAA reduction.

2. Notably, lysine (Lys) levels are nearly doubled following Path knockdown, yet this unexpected increase is not addressed or discussed in the manuscript. Can the authors comment on this finding?

3. Please define nutrient restriction. Is it complete starvation?

4. Is Path expression in the brain restricted to PG cells? How about its expression in the SPG?

5. The title "…SCL36 amino acid transporter Pathetic…" should be "…SLC36 amino acid transporter Pathetic…"

Rev. 2:

Background

Normal development of an animal depends on the availability of nutrients, including amino acids. Upon moderate nutrient deprivation undersized adults form with all organs formed isometrically with body size. However, when nutrient restriction occurs late in development, a preferential growth of the CNS is observed at the expense of other organs. This process has been termed brain sparing and occurs in humans as well as in flies. The Drosophila amino acid transporter pathetic (path) was previously found to be needed for overall body growth and dendrite growth (PMID: 26063572). The Path transporter is a member of the SlC36A family which transports proline, alanine, or tryptophan. A few years ago, path was found to be required for neural stem cell proliferation under nutrient restriction (PMID: 32741363). While Path is expressed in several different glial cell types as well as in neuroblasts, the glial expression domain is responsible for stem cell proliferation under nutrient restriction (PMID: 32938923). The nutrient status also affects tumor growth in a Path dependent manner. When the oncogenic form of Ras, RasG12V is expressed in the eye disc of C-terminal Scr kinase mutant larvae, tumors develop. Strikingly, when such animals are raised on a high-sugar diet tumor formation is enhanced and muscle wasting is observed. The excessive tumor growth and the muscle wasting depends on circulating proline and the Path transporter.

Dong et al.,

During normal larval development previous work of L. Cheng (PMID: 21816278) defined the critical weight period after which brain sparing is observed under nutrient restriction. In the present paper the group now extends these old findings and asks how growth of brain tumor is affected by nutrient restriction.

In wild type larvae, nutrient restriction does not affect brain growth (Fig1D-F, here shown by anti-Mira/pH3 staining). When neuroblast division is triggered by QF/QUAS dependent silencing of prospero, brain tumors develop that are sensitive to nutrient restriction (Fig1G-I, here show by anti-Dpn staining). Importantly, brains stay small. Whether or not brain sparing is affected is not easy extractable from the Figures, since different antibodies are used, and the scale bars are hard to read. The authors then show that reduction in tumor growth is due to a slowdown in cell cycle progression and that nutrient restriction can also affect glial tumor growth, but not epithelial tumors induced by scribble knockdown (note there is a typo the gene is called scribble not scribbled) the eye imaginal disc.

In a next step the authors analyzed whether nutrient restriction affects the number of glial cells and stained larvae for glial nuclei using the Repo marker. In both the tumor model as well as in wild type brain, nutrient restriction results in a reduction of the overall number of glial cells (What are the actual numbers?). While this is clearly documented, no attempts are made to determine the glia cell type affected. The authors only use Fly-FUCCI and a driver that is specific to adult perineurial glia (the expression pattern in the larva is not shown, is there no neuronal expression?) supports the notion that these glial cells are affected but they do not show a specific role of perineurial glia. One must count the numbers of the different glial subtypes in the different settings using the tools that are available.

Next the authors analyzed amino acid transporter expression in fed and nutrient restricted tumor bearing larvae. Under nutrient restriction, three amino acid transporters Minidiscs (MND), Juvenile hormone Inducible-21 (JhI-21) and Pathetic (Path), are transcriptionally downregulated. Where these transporters are expressed is not mentioned. MND is not expressed in the blood-brain barrier, while JhI-21 and path are expressed. MND and Jhl-21 transport branched-chain amino acids (Leu, Ile, and Val) and removal of Leu and Ile after reaching the critical weight resulted in a reduction of tumor size and glial cell number. I failed to see arguments, however, supporting the conclusion that branched-chain amino acids "play a key role in regulating perineurial proliferation and, consequently, tumor growth".

Instead of following up on the role of MND and Jhl-21 the authors turn to an analysis of Path. Path is expressed by glia as well as in neuroblasts (the latter not being mentioned in the paper). Unfortunately, all stainings of Path::GFP are of very low quality that do not allow any conclusions on the effect of nutrient restriction on path expression. In addition, to immune fluorescent imaging data, western blots must be used to demonstrate these claimed expression changes. Why do the images in S5 FigA'/B' look so different to what is shown in S5 FigJ,K.

To further deduce the role of path in perineurial glia the authors performed repo-Gal4 based knockdown experiments. The knockdown efficiency is documented in S6 FigA. Here is a mismatch between the image shown and the quantification shown in S6 FigC. Moreover, the perineurial glia Gal4 driver GMR85G01-Gal4 must be used in this control experiment to judge whether there it can induce any change in path expression.

The statement that nutrient restriction induces a differential regulation of path expression in glia depending on the presence of brain tumors is fascinating - but unfortunately the overall low quality of the images makes it hard to believe. Add western blot experiments.

Gain of function of path is claimed to override the effect of nutrient restriction in glial and tumor growth. What is the control here? Is the effect of Path overexpression the same as observed to what is noted upon Path knockdown?

Quantification of brain amino acid concentrations indicates a decrease in the amount of branched-chain amino acids but not the ones that are normally transported by Path - but which are transported by MND and JhI-21. The interpretation is that Path regulates transport of such amino acids. This is an interesting possible conclusion but instead of addressing it in more details the authors switch to the analysis how Path expression is regulated and analyzed PI3K signaling that eventually leads to a regulation of the nuclear localization of FOXO. Again, I must confess that I was not convinced by the data shown that in nutrient restricted larvae FOXO localizes more to nuclei. Likewise, the analysis as well as the description of the experiment shown in Fig 5E-H is not convincing. The reduction of Path could be judged from the images BUT why is the repo-RFP signal then different in every image? And how can a pan-glial expressed RFP marker serve as surface glia marker? The claim that nutrient restriction downregulates PI3K signaling in perineurial glia is not supported by the data.

---

## [Decision Letter · Decision Letter 2]

6 Oct 2025

Dear Dr Cheng,

Thank you for your patience while we considered your revised manuscript entitled "The blood-brain barrier regulates brain tumour growth specifically via the SCL36 amino acid transporter Pathetic in Drosophila" for publication as a Research Article at PLOS Biology. This revised version of your manuscript has been evaluated by the PLOS Biology editors, the Academic Editor and the two original reviewers.

Based on the reviews, we are likely to accept this manuscript for publication, provided you satisfactorily address the remaining points raised by the reviewers. Regarding Reviewer 1's comments, direct metabolite flux data won't be needed, the changes can be done in writing. Please also make sure to address the data and other policy-related requests stated below my signature.

In addition, we would like you to consider a suggestion to improve the title:

"The blood-brain barrier regulates brain tumor growth through the SCL36 amino acid transporter Pathetic in Drosophila"

We expect to receive your revised manuscript within two weeks.

*Published Peer Review History*

*Press*

Sincerely,

Ines

--

Ines Alvarez-Garcia, PhD

Senior Editor

PLOS Biology

Note that we do not require all raw data. Rather, we ask that ALL INDIVIDUAL QUANTITATIVE OBSERVATIONS that underlie the data summarized in the figures and results of your paper be made available in one of the following forms:

Fig. 1H, I, L, O, R, U, V, Y; Fig. 2G, H, J-M, P, U, V; Fig. 3F-H, M-O, R, S; Fig. 4C, D, G-N; Fig. 5D, H, I, L, M, Q, R, T, U; Fig. 6E, J, M, N, Q, R; Fig. 7F-L, O; Fig. S1C-E, J-L; Fig. S2C, F, I; Fig. S3C, F, I, J, M; Fig. S4D, M; Fig. S5A, B; Fig. S6A-H; Fig. S7C, F, I, L; Fig. S8C, D, G, J, K, N, O, R, U, V; Fig. S9A-R and Fig. S10A, D, G-K

CODE POLICY

Reviewers' comments

Rev. 1:

The authors have satisfactorily addressed nearly all reviewer comments. One point remains unresolved regarding the suggestion to demonstrate that Path is a BCAA transporter. In the absence of direct metabolite flux data, it is recommended that the authors consider an alternative interpretation that Path may function as a transceptor (https://pmc.ncbi.nlm.nih.gov/articles/PMC4470281/) rather than a transporter, and that the observed changes in amino acid levels could be mediated by amino acid-binding events causing rather than transport per se.

Rev. 2:

The authors revised the manuscript according to all essential editorial comments and also addressed all other concerns of both reviewers.

I have only two minor points:

- In Figure 1G, the yellow boxing appears displaced.

- Please emntion the number of counted glial cells in the main text.

---

## [Editor Report · Decision Letter 3]

30 Oct 2025

Dear Dr Cheng,

Thank you for the submission of your revised Research Article entitled "The blood-brain barrier regulates brain tumour growth through the SLC36 amino acid transporter Pathetic in Drosophila" for publication in PLOS Biology. On behalf of my colleagues and the Academic Editor, Richard Daneman, I am delighted to let you know that we can in principle accept your manuscript for publication, provided you address any remaining formatting and reporting issues. These will be detailed in an email you should receive within 2-3 business days from our colleagues in the journal operations team; no action is required from you until then. Please note that we will not be able to formally accept your manuscript and schedule it for publication until you have completed any requested changes.

PRESS

Sincerely, 

Ines

--

Ines Alvarez-Garcia, PhD

Senior Editor

PLOS Biology
